# Target cell-specific synaptic dynamics of excitatory to inhibitory neuron connections in supragranular layers of human neocortex

Mean-Hwan Kim[1]*, Cristina Radaelli[1], Elliot R Thomsen[1], Deja Monet[1], Thomas Chartrand[1], Nikolas L Jorstad[1], Joseph T Mahoney[1], Michael J Taormina[1], Brian Long[1], Katherine Baker[1], Trygve E Bakken[1], Luke Campagnola[1], Tamara Casper[1], Michael Clark[1], Nick Dee[1], Florence D'Orazi[1], Clare Gamlin[1], Brian E Kalmbach[1,2], Sara Kebede[1], Brian R Lee[1], Lindsay Ng[1], Jessica Trinh[1], Charles Cobbs[3], Ryder P Gwinn[3], C Dirk Keene[4], Andrew L Ko[5], Jeffrey G Ojemann[5], Daniel L Silbergeld[5], Staci A Sorensen[1], Jim Berg[1], Kimberly A Smith[1], Philip R Nicovich[1], Tim Jarsky[1], Hongkui Zeng[1], Jonathan T Ting[1,2], Boaz P Levi[1], Ed Lein[1,4,5]

[1]Allen Institute for Brain Science, Seattle, United States; [2]Department of Physiology & Biophysics, School of Medicine, University of Washington, Seattle, United States; [3]Swedish Neuroscience Institute, Seattle, United States; [4]Department of Laboratory Medicine & Pathology, School of Medicine, University of Washington, Seattle, United States; [5]Department of Neurological Surgery, School of Medicine, University of Washington, Seattle, United States

**\*For correspondence:**
meanhwank@alleninstitute.org

**Abstract** Rodent studies have demonstrated that synaptic dynamics from excitatory to inhibitory neuron types are often dependent on the target cell type. However, these target cell-specific properties have not been well investigated in human cortex, where there are major technical challenges in reliably obtaining healthy tissue, conducting multiple patch-clamp recordings on inhibitory cell types, and identifying those cell types. Here, we take advantage of newly developed methods for human neurosurgical tissue analysis with multiple patch-clamp recordings, *post-hoc* fluorescent in situ hybridization (FISH), machine learning-based cell type classification and prospective GABAergic AAV-based labeling to investigate synaptic properties between pyramidal neurons and PVALB- vs. SST-positive interneurons. We find that there are robust molecular differences in synapse-associated genes between these neuron types, and that individual presynaptic pyramidal neurons evoke postsynaptic responses with heterogeneous synaptic dynamics in different postsynaptic cell types. Using molecular identification with FISH and classifiers based on transcriptomically identified PVALB neurons analyzed by Patch-seq, we find that PVALB neurons typically show depressing synaptic characteristics, whereas other interneuron types including SST-positive neurons show facilitating characteristics. Together, these data support the existence of target cell-specific synaptic properties in human cortex that are similar to rodent, thereby indicating evolutionary conservation of local circuit connectivity motifs from excitatory to inhibitory neurons and their synaptic dynamics.

## Editor's evaluation

The authors have made paired recordings from synaptically connected excitatory and inhibitory neurons in slices of human neocortex and used posthoc molecular methods to identify major

subclasses of the recorded interneurons. The principal finding is that as found previously in the rodent cortex, the short-term plasticity of the synaptic connections from excitatory to inhibitory neurons depends on the molecular identity of the inhibitory neurons. Hence an important functional principle of connectivity is conserved.

## Introduction

Synaptic transmission is a fundamental means to convey information between neurons and can be modulated by many factors including the intrinsic membrane properties of pre- and postsynaptic cell types, their connection probability, location of synapses, and synaptic short-term plasticity (STP) with timescales from milliseconds to minutes. Diverse forms of STP exist that involve differences in presynaptic release probability of neurotransmitters, calcium accumulation in presynaptic terminals, and retrograde signaling from postsynaptic dendrites with rapid timescales (*Abbott and Regehr, 2004*). Importantly, the properties of individual synapses from a given neuron are often determined by the identity of the postsynaptic neurons. Target cell-specific short-term synaptic dynamics from excitatory to inhibitory neuron connections have been identified in many brain regions including neocortex, cerebellum, and hippocampus (*Blackman et al., 2013*).

Rodent studies from multiple brain regions have begun to elucidate differential synaptic properties between specific neuron types, as well as their underlying postsynaptic molecular mechanisms. For example, specific postsynaptic molecules controlling presynaptic transmitter release have been identified, including *Elfn1* (extracellular leucine rich repeat and fibronectin Type III domain containing 1) (*Sylwestrak and Ghosh, 2012*), N-cadherin and β-catenin (*Vitureira et al., 2011*), PSD-95-neuroligin (*Futai et al., 2007*) in hippocampus, and *Munc13-3* (*Augustin et al., 2001*) in cerebellum. In cerebral cortex, excitatory to morphologically defined multipolar basket cell synapses show a high initial release probability and synaptic depression. The GABAergic inhibitory interneuron basket cells are known to express the gene parvalbumin (PVALB); therefore, we will use the term PVALB interneurons to describe them, and use typical convention to refer to expression of the parvalbumin gene as *PVALB* for mRNA and PVALB for protein in human, and *Pvalb* for mRNA and PVALB for protein in rodent. In contrast, excitatory to morphologically defined bi-tufted (or low threshold activated, somatostatin-positive or SST interneurons) cell synapses show low initial release probabilities and synaptic facilitation (*Reyes et al., 1998*; *Koester and Johnston, 2005*). This specialized short-term facilitation in SST interneurons is known to be mediated by *Elfn1* expression in postsynaptic dendritic shafts of SST cells (*Sylwestrak and Ghosh, 2012*; *de Wit and Ghosh, 2016*; *Stachniak et al., 2019*), but not in PVALB neurons. This molecular mechanism was originally discovered in the hippocampus but was extended to the cerebral cortex showing that *Elfn1* in postsynaptic SST neurons interacts with presynaptic metabotropic glutamate receptors (mGluRs) and kainate receptors in a layer-specific manner (*Stachniak et al., 2019*).

Addressing whether similar synaptic properties and molecular mechanisms are conserved in human cortex has been extremely challenging due to limitations in tissue access and available methods. Advances in single cell transcriptomics have demonstrated a highly complex cellular architecture in the mouse and human cortex, with 45 types of inhibitory interneurons reported that can be discriminated using genome-wide gene expression (*Hodge et al., 2019*). These neuron types are organized hierarchically, with levels referred to as class, subclass, and type. PVALB and SST neurons correspond to major divisions among GABAergic interneurons at the subclass level, along with LAMP5 and VIP subclasses. Comparative transcriptomic analyses show a generally conserved cell type organization from mouse to human, but with many changes in cellular gene expression that suggest differences in cellular physiology, anatomy, and connectivity (*Hodge et al., 2019*; *Bakken et al., 2021*). Whether these species differences lead to functional differences has been a topic of great debate, as it is well known that the same functional readouts, such as synaptic connectivity and dynamics, could be achieved through distinct molecular mechanisms across species (*Goaillard and Marder, 2021*). Recent work has shown that electrophysiological properties and local synaptic connectivity can be studied in acute human neocortical slices derived from surgical resections (*Molnár et al., 2008*; *Jiang et al., 2012*; *Testa-Silva et al., 2014*; *Kalmbach et al., 2018*; *Beaulieu-Laroche et al., 2018*; *Boldog et al., 2018*; *Seeman et al., 2018*; *Peng et al., 2019*; *Planert et al., 2021*; *Campagnola et al., 2021*). These studies have demonstrated many conserved features, but also a variety of human specializations compared to rodents including faster

recovery from synaptic depression (*Testa-Silva et al., 2014*) and greater numbers of functional release sites (*Szegedi et al., 2016*). Thus, conservation of cellular properties between human and model organisms is often seen but cannot be assumed, and it is important to directly compare these properties to understand how well other organisms effectively model the human condition.

The current study aimed to determine whether the target cell-dependent synaptic properties between excitatory pyramidal neurons and PVALB interneurons vs. SST or other non-PVALB interneurons seen in rodent are conserved in human. We leveraged a number of technological advances to address this question, including (1) multiple patch-clamp recordings to analyze intrinsic membrane properties and local synaptic connectivity and STP, (2) *post-hoc* multiplexed fluorescent in situ hybridization (mFISH) to reveal molecular properties of synaptically connected neurons, (3) a human slice culture approach with cell class-specific adeno-associated virus (AAV) vectors to prospectively label GABAergic interneurons with fluorescent reporter genes, and (4) a machine-learning-based classifier to predict interneuron subclass identity based on a training set of human Patch-seq data that characterized cellular morphology and electrophysiological response properties in transcriptomically identified neurons (*Lee et al., 2021*; *Lee et al., 2022*). We find that STP in human cortex is target cell-specific. Excitatory to fast spiking (or PVALB) synapses show a high initial release probability and synaptic depression, whereas a subset of postsynaptic neurons with facilitating synapses showed *SST* expression by mFISH. Expression of *ELFN1* in human cortex is restricted to non-PVALB types similar to observations made in mouse, suggesting a conservation of molecular machinery mediating these target cell-specific synaptic properties.

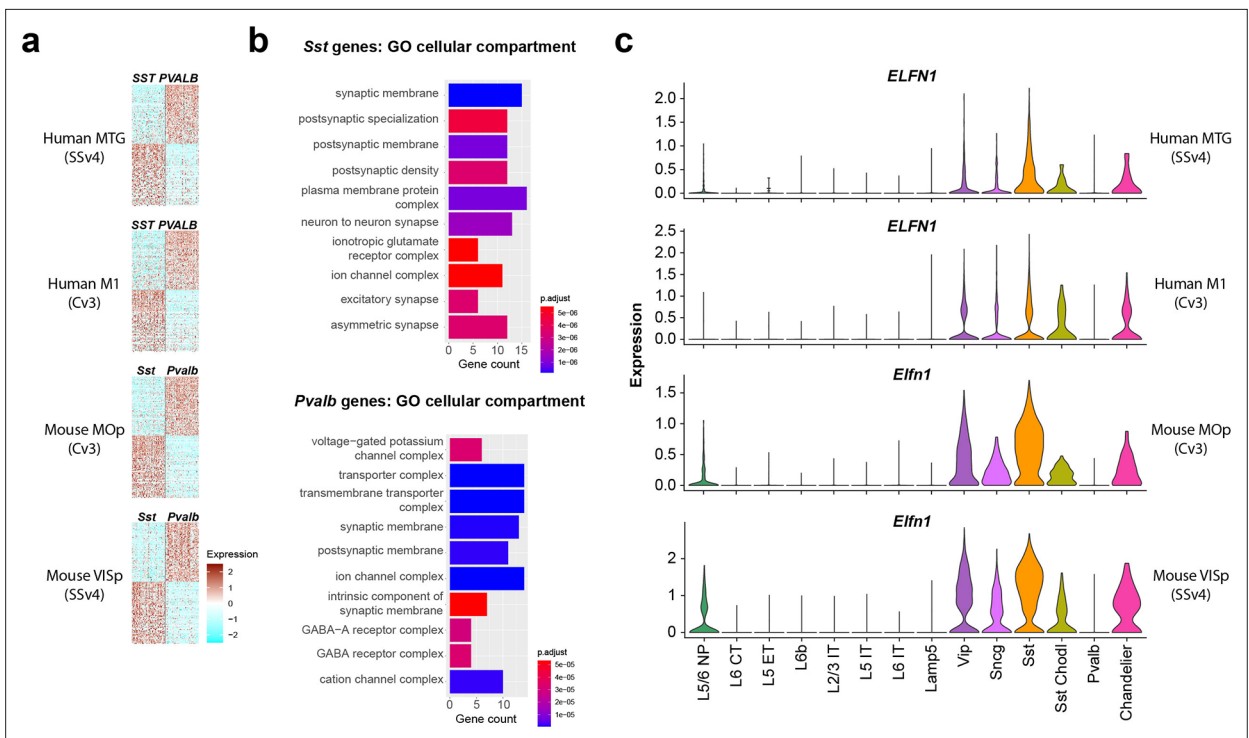

**Figure 1.** Single-nucleus transcriptomic differences between PVALB and SST types in human and mouse cortex. (**a**) Heatmaps showing scaled log2 normalized expression of 147 differentially expressed genes (DEGs) that distinguish PVALB and SST types in both human MTG and mouse VISp. These genes show similar specificity in human M1 and mouse MOp, indicating conserved patterning across cortical areas. Heatmaps show 100 randomly sampled nuclei from each type. SSv4 indicates SMARTseq V4 chemistry, and Cv3 indicates 10 x Chromium V3 chemistry. (**b**) Gene ontology analysis for cellular compartment using conserved DEGs between SST and PVALB. Top 10 enriched categories are involved in synaptic structure and function. (**c**). Violin plots showing neuronal subclass expression levels of *ELFN1* in human and *Elfn1* in mouse cortex, illustrating selective expression in non-PVALB inhibitory neuron subclasses.

## Results

### Conserved GABAergic interneuron subclass gene expression from mouse to human

Single-nucleus transcriptomic analyses from the neocortex of various species have identified a hierarchical classification of neuronal cell types that is conserved across cortical regions and species (*Hodge et al., 2019*; *Bakken et al., 2021*). This classification is consistent with a large literature describing stereotyped anatomy, physiology, and connectivity, for instance for the major subclasses of cortical GABAergic neurons (e.g. PVALB, SST, and VIP) (*Paul et al., 2017*; *Huang and Paul, 2019*). Importantly, transcriptomic analysis of the GABAergic subclasses in mouse cortex shows they are well differentiated from one another by genes involved in synaptic communication (*Paul et al., 2017*; *Huang and Paul, 2019*; *Smith et al., 2019*), suggesting a molecular substrate for their distinctive features of functional synaptic communication.

A comparison of the conservation and divergence of cell subclass markers was performed previously in *Bakken et al., 2021*, analyzing markers of each cortical GABAergic interneuron subclass versus all other GABAergic interneuron subclasses combined. This showed that a core set of markers was robustly specific and conserved, although surprisingly most subclass-selective markers were not. In this study we reanalyzed these data to specifically looked genes that differentiated PVALB versus SST neurons in both mouse and human cortex, as a potential substrate for conserved functional properties. We identified 72 PVALB- and 75 SST-enriched genes whose expression patterns were conserved in both human medial temporal gyrus (MTG) and mouse primary visual cortex (VISp; *Tasic et al., 2018*; *Hodge et al., 2019*; *Figure 1a*, *Supplementary file 1*). These patterns were similar in primary motor cortex (M1 in human, MOp in mouse), consistent with reports of similar transcriptomic GABAergic neuron type properties across mouse cortical areas (*Tasic et al., 2018*). Next, to understand what the function of this gene set was, we performed Gene ontology (GO) analysis to reveal molecular pathways enriched in PVALB or SST subclasses (*Figure 1b*, top 10 categories by p-value). GO analysis can reveal if a gene set contains higher than expected number of genes associated with a cellular function or a subcellular compartment. The most significantly enriched GO terms were for synapse related categories, with postsynaptic membrane Gene Ontology term GO:0045211 (http://www.informatics.jax.org/vocab/gene_ontology/GO:0045211) being enriched in both PVALB and SST neurons. This suggests that many of the conserved genes that discriminate between PVALB and SST subclass cells are involved in regulating synaptic connectivity and their functional properties. Among these, the *Elfn1* gene, described to mediate selective short-term facilitation in SST interneurons (*Sylwestrak and Ghosh, 2012*; *de Wit and Ghosh, 2016*; *Stachniak et al., 2019*), is enriched in GABAergic interneurons compared to excitatory neurons, and in SST and all other GABAergic subclasses except PVALB interneurons, we found that this pattern is conserved across both species and cortical areas (*Figure 1c*). Together, these analyses suggest there are molecular substrates for conserved synaptic properties that differentiate PVALB and SST subclass interneurons (*Blackman et al., 2013*).

### Local synaptic connectivity and intrinsic membrane properties in acute and virally transduced neurons from human ex vivo cultured cortical slices

Next, we investigated local synaptic dynamics from excitatory to inhibitory neurons in the supragranular layer of human neocortex using multiple patch-clamp recordings. In this study, neocortical tissues from 31 donors were used for data collection, derived from neurosurgical resections to treat intractable epilepsy (n=15 cases, n=59 connected pairs) or remove deep brain tumors (n=16 cases, n=30 connected pairs). These tissues were distal to the epileptic focus or tumor, and have shown minimal pathology when examined (*Berg et al., 2021*). Brain pathology was evaluated using six histological markers that were independently scored by three pathologists. Surgically resected tissues have been used extensively to characterize human cortical physiology and anatomy (*Berg et al., 2021*).

Two main experimental approaches were applied including acute brain slice preparation and organotypic brain slice culture preparation (*Figure 2*). Donors included males and females across adult ages (18, minimum age; 80, maximum age; average age 43.45±17.99 (mean ± s.d.)), and tissues were obtained from both left and right hemispheres and primarily temporal and frontal cortices (*Figure 2—figure supplement 1*). Notably, both applications were typically performed on the same

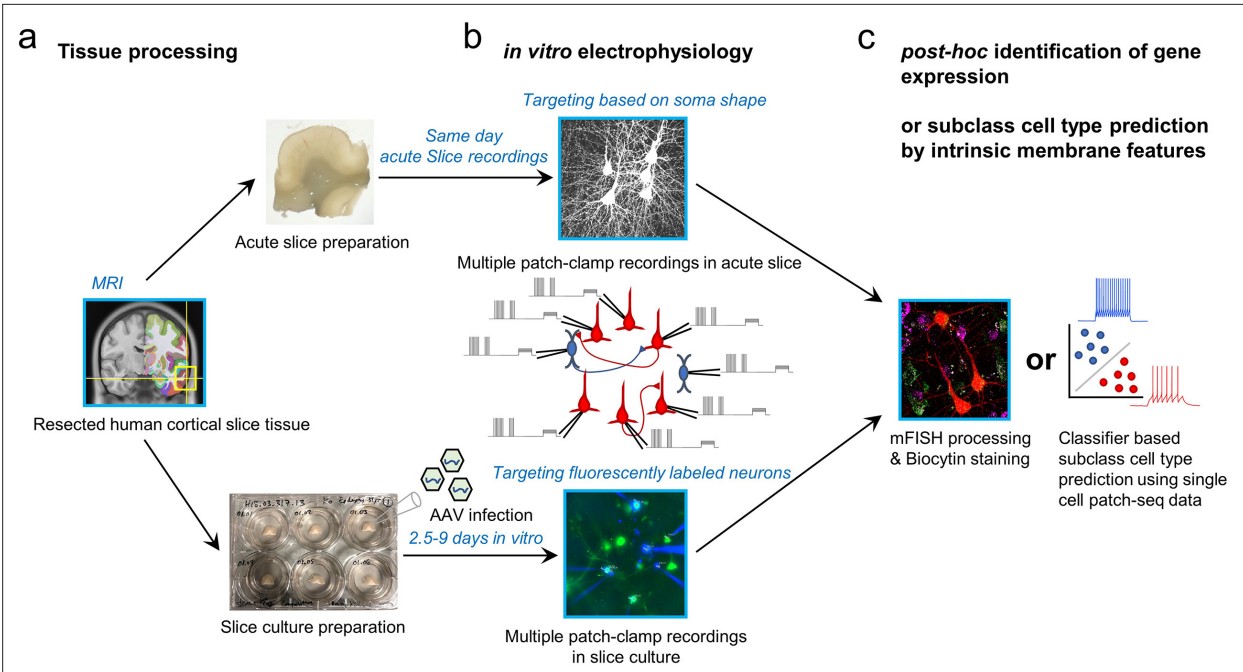

**a** Tissue processing

**b** *in vitro* electrophysiology

**c** *post-hoc* identification of gene expression

or subclass cell type prediction by intrinsic membrane features

*MRI*

Resected human cortical slice tissue

Acute slice preparation

*Same day acute Slice recordings*

*Targeting based on soma shape*

Multiple patch-clamp recordings in acute slice

mFISH processing & Biocytin staining

**or**

Classifier based subclass cell type prediction using single cell patch-seq data

Slice culture preparation

AAV infection *2.5-9 days in vitro*

*Targeting fluorescently labeled neurons*

Multiple patch-clamp recordings in slice culture

**Figure 2.** Schematic of experimental workflow. (**a**) Human neocortical tissue from neurosurgical resections in either acute slice preparations within 45 min following scalpel excision from the patient (upper) or organotypic slice culture preparation with viral transduction (lower). (**b**) Up to eight simultaneous patch-clamp recordings are performed on either acute slices (upper) or slice culture after 2.5–9 days in vitro (lower). Targeting of neurons is either carried out by visually identifying cell bodies using an upright microscope with oblique illumination (upper) or by targeting neurons expressing fluorescent reporters following viral transduction (lower). (**c**) To identify subclass cell types in connectivity-assayed neurons, we applied multiplexed fluorescence in situ hybridization (mFISH) on fixed slices to identify marker gene expression, or used a machine learning classifier with cellular intrinsic membrane properties measured after connectivity assays.

The online version of this article includes the following figure supplement(s) for figure 2:

**Figure supplement 1.** Summary of donor information.

surgical cases, since multiple slices could be generated from these resections whose average volume was 1.39±0.57 cm³ (mean ± nders had no role in study dstandard error of mean (s.e.m); averaged over n=12 cases). Acute experiments were performed within 12 hr following surgical resection, whereas slice culture experiments were performed between 2.5 and 9 days in vitro (DIV; *Figure 2*). In acute slice preparations, neurons were targeted based on somatic shape as visualized by oblique illumination. In slice culture experiments, AAV vectors were used to transduce GABAergic neurons and drive expression of fluorescent reporters under the control of cell-class-selective regulatory elements to facilitate targeting labeled neurons for multiple patch-clamp recordings (*Schwarz et al., 2019*; *Mich et al., 2021*). The use of reporter AAV vectors can greatly facilitate prospective marking of GABAergic cells, and has not been seen to significantly impact the physiology or morphology of cortical neurons (*Lee et al., 2022*). After multiple patch-clamp electrophysiology experiments, two strategies were used to identify GABAergic subclass identity, including an mFISH analysis with subclass markers and a machine learning classifier based on Patch-seq analysis using similar human slice preparations.

Connectivity assays with multiple patch-clamp recordings were performed by targeting cell bodies located between 50 and 120 µm below the surface of the slice to minimize truncation of dendrites and other superficial damage that occurs during slice preparation (*Figure 3a–g*; *Seeman et al., 2018*; *Campagnola et al., 2021*). To look at synaptic connections between excitatory pyramidal cells and inhibitory interneurons in acute experiments, we simultaneously targeted cells with pyramidal shape and cells with small, round somas (putative interneurons). For example, *Figure 3* shows a multiple patch-clamp recording that successfully targeted 4 pyramidal neurons (cell morphology reconstruction after biocytin labeling; *Figure 3i*) and one interneuron. Connectivity was observed from excitatory to inhibitory interneuron, which displayed fast spiking characteristics (cell 4) and a distinct spike shape compared to pyramidal neurons (*Figure 3e and g*), with strong excitatory postsynaptic potential

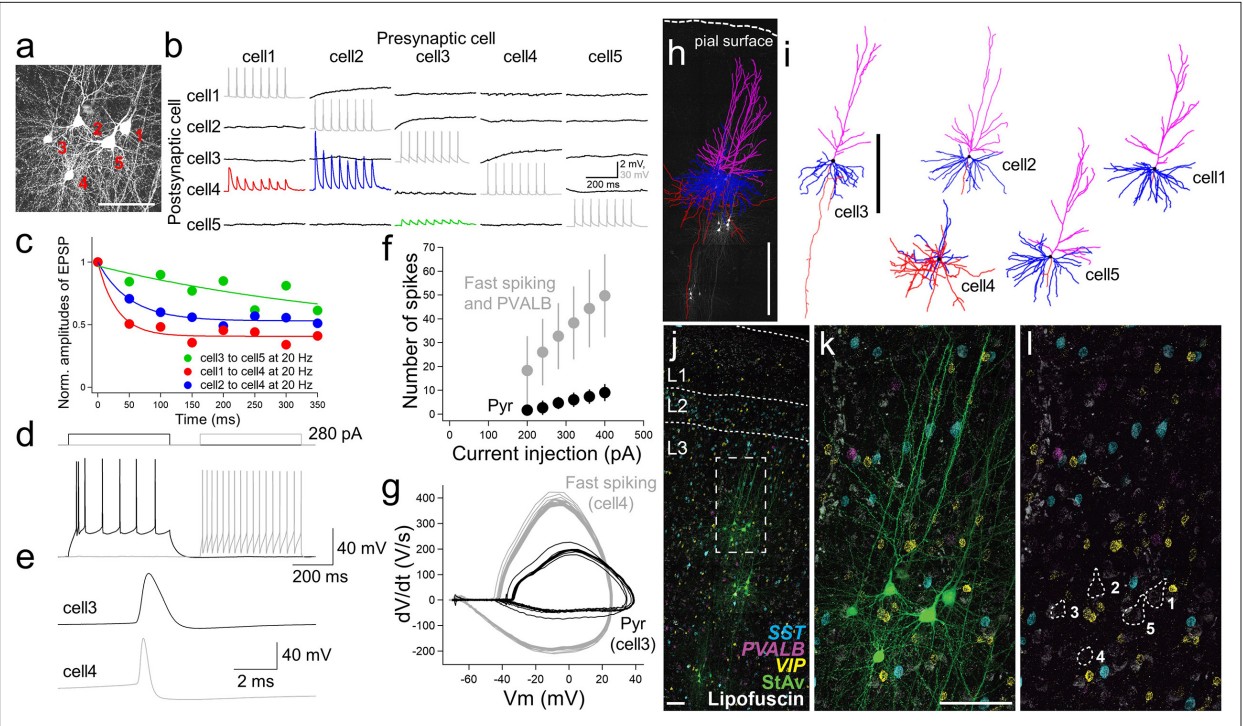

**Figure 3.** Physiology, morphology, and molecular identification of pyramidal-interneuron connections in human neocortex. Example experiment using acute slice preparation with five cells simultaneously patched. (**a**) Maximum intensity projection montage confocal image of biocytin/streptavidin labeling. Scale bar, 50 μm. (**b**) Corresponding membrane voltage traces from connectivity assay. Presynaptic action potentials (gray) in individual neurons (cell 1 to cell 5) were sequentially generated by 8 brief current pulses at 20 Hz while simultaneously recording the postsynaptic membrane voltage in non-stimulated neurons in current-clamp mode (black). Traces averaged over 10 repetitive 8 pulse stimulations. This probing uncovered a strong and adapting excitatory synaptic connection from cell 2 to cell 4 (blue trace) and cell 1 to cell 4 (red trace) compared to the synaptic connection from cell 3 to cell 5 (green). (**c**) Summary plot of short-term synaptic dynamics with presynaptic 20 Hz stimulation (8 pulses at 20 Hz) in connected pairs as in (**b**). Amplitude normalized to size of initial EPSP. (**d**) Example traces of action potential generation by step current injection in regular spiking (cell 3, black) and fast spiking neurons (cell 4, gray). The same amount of current injection (280 pA) was applied to cell 3 and cell 4.(**e**) Spike shape comparison between regular and fast spiking neurons detected in the connectivity assay shown in (**b**). (**f**) Frequency-current curve of pyramidal neuron (Pyr; mean ± standard deviation, n=3), and fast spiking neuron (panel (**k**), cell 4) and PVALB positive neurons (including upper 2 cells shown in panel (**g**) of *Figure 3— figure supplement 1*) (mean ± standard deviation, n=3). (**g**), Phase plot (dV/dt vs V) analysis based on responses shown in (**d**). (**h**) Morphological reconstruction of the 5 recorded neurons shown in (**a**) with biocytin +confocal imaging. Scale bar, 500 μm.(**i**) Reconstruction of individual neurons. Scale bar, 500 μm. Blue, magenta, and red indicate basal dendrites, apical dendrites, and axons in pyramidal neurons (cell 1,2,3,5). For the interneuron (cell 4), blue and red indicate dendritic and axonal structures, respectively. (**j**) Fluorescence montage of cells imaged in (**a**), (**j-l**) stained by mFISH for inhibitory neuron subclass markers (*PVALB*, *SST*, and *VIP*) and biocytin. Multiple patch-clamp recordings were performed on three separate cell clusters in this slice (**j**). Note, substantial lipofuscin is observed in this slice. White box in (**j**) is shown at higher magnification for mFISH and biocytin (**k**), or mFISH only (**l**).

The online version of this article includes the following figure supplement(s) for figure 3:

**Figure supplement 1.** Two examples of multiple patch-clamp recordings with excitatory and inhibitory HCR markers in acute ex vivo human neocortex and depth dependence of HCR signals.

**Figure supplement 2.** Examples of multiple patch-clamp recordings with inhibitory interneuron-targeted HCR staining in organotypic slice cultures.

**Figure supplement 3.** HCR signal comparison between patched and unpatched neighboring neurons.

(EPSP) responses that rapidly depressed (e.g. cell 1 to cell 4 and cell 2 to cell 4). In contrast, the responses between connected pyramidal neurons were small with weakly depressing characteristics (e.g. cell 3 to cell 5; *Figure 3b and c*). The intrinsic membrane features (*Figure 3d–g*) and morphology (*Figure 3h and i*) of this interneuron were consistent with the identity of a PVALB cell (*Reyes et al., 1998*).

We used hybridization chain reaction (HCR) mFISH to identify postsynaptic interneuron subclass identities following human multiple patch-clamp experiments. This method was used because it penetrates tissue efficiently (*Choi et al., 2010*), allows strong signal amplification, has high signal-to-noise with background-reducing probe design (*Choi et al., 2018*), and allows multiple rounds of stripping

and re-probing (*Figure 3—figure supplements 1–3*). Following multiple patch-clamp recordings, slices were fixed, passively cleared, and stained by mFISH using HCR kit version 3.0 (*Shah et al., 2016*; *Choi et al., 2018*). Messenger RNA from prominent excitatory (*SLC17A7*, solute carrier family 17 member 17, also known as Vesicular Glutamate Transporter 1; *Aihara et al., 2000*) and inhibitory (*GAD1,* glutamic acid decarboxylase 1) marker genes were easily resolved in both patched (biocytin/ streptavidin, StAv) and neighboring non-patched neurons (*Figure 3—figure supplement 1a–d*). As expected, *SLC17A7* and *GAD1* expression was mutually exclusive in excitatory and inhibitory neurons, respectively, and only GAD1+ cells were found in layer 1. *SLC17A7* and *GAD1* mRNA staining was comparable between patched and neighboring non-patched neurons after long whole-cell record-ings (around 30–75 min; *Figure 3—figure supplement 1b, c, i and j*). We were also able to resolve *SLC17A7* and *GAD1* mRNA staining through the depth of the slice, and did not observe changes of averaged fluorescent intensities in individual neurons by depth (*Figure 3—figure supplement 1j and k*). In an acute slice, no noticeable change was observed between patched and non-patched SLC17A7+ cells, but a single patched GAD1 + cell showed lower HCR signal than neighboring GAD1 + cells (*Figure 3—figure supplement 1*). Alternatively, in a cultured slice, expression of *PVALB* was reduced in patched cells relative to unpatched neighbors (*Figure 3—figure supplements 2 and 3*). The ability to stain for marker genes across multiple rounds allowed probing for an increased number of genes, and re-probing for genes that produced low signal from the first round such as *VIP* (*Figure 3—figure supplement 2c–d* cell 3). One challenge to this approach is that human brain tissue often exhibits dense lipofuscin around some somatic structures that is highly autofluorescent and causes challenges with detecting fluorescent signals in mFISH. Lipofuscin is a cellular metabolic byproduct that accu-mulates with age in lysosomes, and is also known as age pigment (*Boellaard and Schlote, 1986*; *Figure 3j–l*, *Figure 3—figure supplement 1e–g*), that persists after tissue clearing with 8% SDS and throughout the mFISH staining procedure. However, it was possible to distinguish the distribution of amplified mRNA fluorescent dots from lipofuscin autofluorescence by imaging across multiple fluo-rescent channels, as lipofuscin produced fluorescence in all channels (e.g. *Figure 3—figure supple-ment 1e–g*). *PVALB* labeling could be detected, but the staining was often very weak. We observed this with several patched *PVALB*-positive cells, where *PVALB* mRNA abundance was at lower levels than adjacent unpatched PVALB + cells (*Figure 3—figure supplement 1e–g*). Whether this reflects real differences in mRNA abundance between cells, or loss of mRNA during multiple patch-clamp recording is unclear.

To efficiently target GABAergic interneurons for multiple patch-clamp recordings, we also performed rapid viral genetic labeling of cortical GABAergic interneurons in human organotypic slice cultures (*Figure 3—figure supplement 2a and b*; see Methods; *Ting et al., 2018*; *Mich et al., 2021*). We used an adeno-associated virus (AAV), with capsid PHP.eB that drives SYFP2 reporter expression under the control of an optimized version of a previously described forebrain GABAergic neuron enhancer (*Stühmer et al., 2002*; *Dimidschstein et al., 2016*; *Mich et al., 2021*). This AAV-DLX2.0-SYFP2 virus was directly applied to the slice surface at a concentration of 1-5e$^{10}$ vg/slice. This virus produced rapid reporter expression, often visible within 2 days, and allowed us to execute physiology experiments as early as 2.5 days in vitro (DIV) after viral administration. We performed targeted multiple patch-clamp recordings of labeled neurons in addition to pyramidal shaped neurons in human cortical slices (*Figure 3—figure supplements 1 and 2*). During the recordings, we observed some differences between multiple patch-clamp recordings in viral labeled slice culture and acute slice preparation. First, giga-ohm seals were more readily obtained between patch pipette and cell membrane in neurons from ex vivo cultured slices compared to acute slices. Second, the somatic structure of unlabeled neurons was more difficult to resolve in slice culture with minimal positive pressure on the patch pipette, making patching unlabeled neurons more challenging. Third, we compared the expression of several marker genes in patched or neighboring unpatched cells and saw only dim expression of *PVALB*, while other genes showed similar expression levels (*Figure 3—figure supplement 3*). This finding is consistent with *PVALB* being difficult to resolve by mFISH after patching. Nonetheless, the ability to exclusively target genetically labeled GABAergic neuronal subclasses in the human neocortex greatly improved the efficiency of targeted recording experiments.

# Diverse synaptic dynamics from excitatory to inhibitory neurons

To analyze excitatory postsynaptic potential (EPSP) dynamics, we stimulated presynaptic neurons with spike trains of 8 pulses at 20 and 50 Hz (see Methods, *Figure 4a*, *Seeman et al., 2018*; *Campagnola et al., 2021*). Recovery from synaptic depression was measured by probing with an additional four pulses after variable inter-spike intervals (62.5ms, 125ms, 250ms, 500ms, 1 s, 2 s and 4 s) following induction by the 8 pulses spike train at 50 Hz in each 15 s interstimulus interval. In our multiple patch-clamp recordings, up to 8 neurons were targeted to patch simultaneously including both pyramidal neurons and interneurons. Therefore, we were able to include many recordings in our analysis such as two presynaptic pyramidal neurons and one connected postsynaptic interneuron (*Figure 4b and c*), or one pyramidal neuron and two connected postsynaptic interneurons (*Figure 4d*). As shown in *Figure 4b and c*, when two pyramidal neurons were connected to the same postsynaptic inhibitory neuron, they typically showed similar kinetics of short-term synaptic plasticity that was either depressing or facilitating depending on the postsynaptic neuron. Similarly, when one presynaptic pyramidal neuron was connected to 2 interneurons, the short-term synaptic plasticity was often different for the two interneurons (*Figure 4d*). Both results indicate that postsynaptic cell identity is a determinant of short-term synaptic dynamics (i.e. target cell-specific) in excitatory to inhibitory neuron connections in human cortex (*Reyes et al., 1998*; *Markram et al., 1998*; *Koester and Johnston, 2005*). These target-dependent synaptic properties were observed in both acute and slice culture preparations (*Figure 4b–d*).

Synaptic dynamics of connected excitatory to inhibitory neuron pairs were analyzed from both acute (n=33 at 50 Hz; n=29 at 20 Hz stimulation protocol) and slice culture preparations (n=56 at 50 Hz; n=52 at 20 Hz stimulation protocol). To quantify synaptic dynamics, initial EPSP amplitudes in each pair of excitatory to inhibitory neuron connections were normalized and displayed as heatmaps (*Figure 4e–h*). Rates of postsynaptic facilitation and depression are usually presynaptic stimulus frequency dependent (*Beierlein et al., 2003*). Here, 50 Hz stimulation protocol showed stronger depression with bigger EPSP responses (upper heatmap in both acute and slice culture data, *Figure 4e and f*) compared to the 20 Hz stimulation protocol (*Figure 4g and h*). These results show that amounts of depression are dependent on presynaptic stimulus frequencies (*Figure 4—figure supplement 1a and b*). To further explore the potential relationship between synapse types and their properties of short-term plasticity, we first defined synapses as facilitating or depressing based on the ratio of EPSP amplitude between the 1st and 2nd pulse (1:2 ratio; as known as paired-pulse ratio, or PPR; *Figure 4k*). Since PPR does not capture the full range of dynamics, we also defined additional metrics such as a 1:8 pulse ratio and a 1:6–8 pulse ratio defined by ratio between first and average of 6th to 8th pulses (*Figure 4—figure supplement 1a and b*; *Varela et al., 1997*; *Tsodyks and Markram, 1997*; *Beierlein et al., 2003*). Using PPR, averaged EPSP kinetics of depressing and facilitating synapses in terms of their rise time and decay (*Figure 4i*) were not statistically different (Wilcoxon rank sum test, *Figure 4—figure supplement 2*). However, the depressing synapses had higher EPSP amplitudes than facilitating synapses (*Figure 4j*, left panel; Linear regression between PPR and EPSP amplitude, R-squared=0.079452, p=0.0074488), and this difference was not accounted for by disease indication or slice preparation method (p=0.00029819 for depression/facilitation, p=0.7852 for epilepsy/tumor, p=0.2766 for acute/culture, Kolmogorov-Smirnov test). When we parameterized the degree of facilitation/depression (i.e. PPR) and recovery rate in early time periods within 250ms, there was a significant correlation between these parameters (*Figure 4—figure supplement 3*) that showed distinct synaptic recovery dynamics between the two synapse types. Their recovery responses (i.e. 9th pulse at various time intervals) were also not accounted for by disease indication (p=0.4781) or slice preparation method (p=0.7816, two-way ANOVA with repeated measures as different recovery time periods). Similarly, normalized synaptic dynamics (i.e. normalized responses from first to 8th pulses at both 20 Hz and 50 Hz; *Figure 4—figure supplement 1c*) were not impacted by disease state of the donor, or slice preparation method with two-way ANOVA with repeated measures (both p>0.05). However, when normalized pulse responses were individually compared, some of them were separated from acute to slice culture preparations (2nd to 6th pulse responses show p<0.05 with false discovery rate (FDR) corrected Wilcoxon rank sum test, at 50 Hz, left panel of *Figure 4—figure supplement 1d*), although we did not observe different dynamics responses related to disease indication (p>0.05 with FDR corrected Wilcoxon rank sum test). Indeed, more facilitating synapses were detected in slice cultures than in acute slices. Based on the train-induced STP (1:6–8 ratio), about 30%

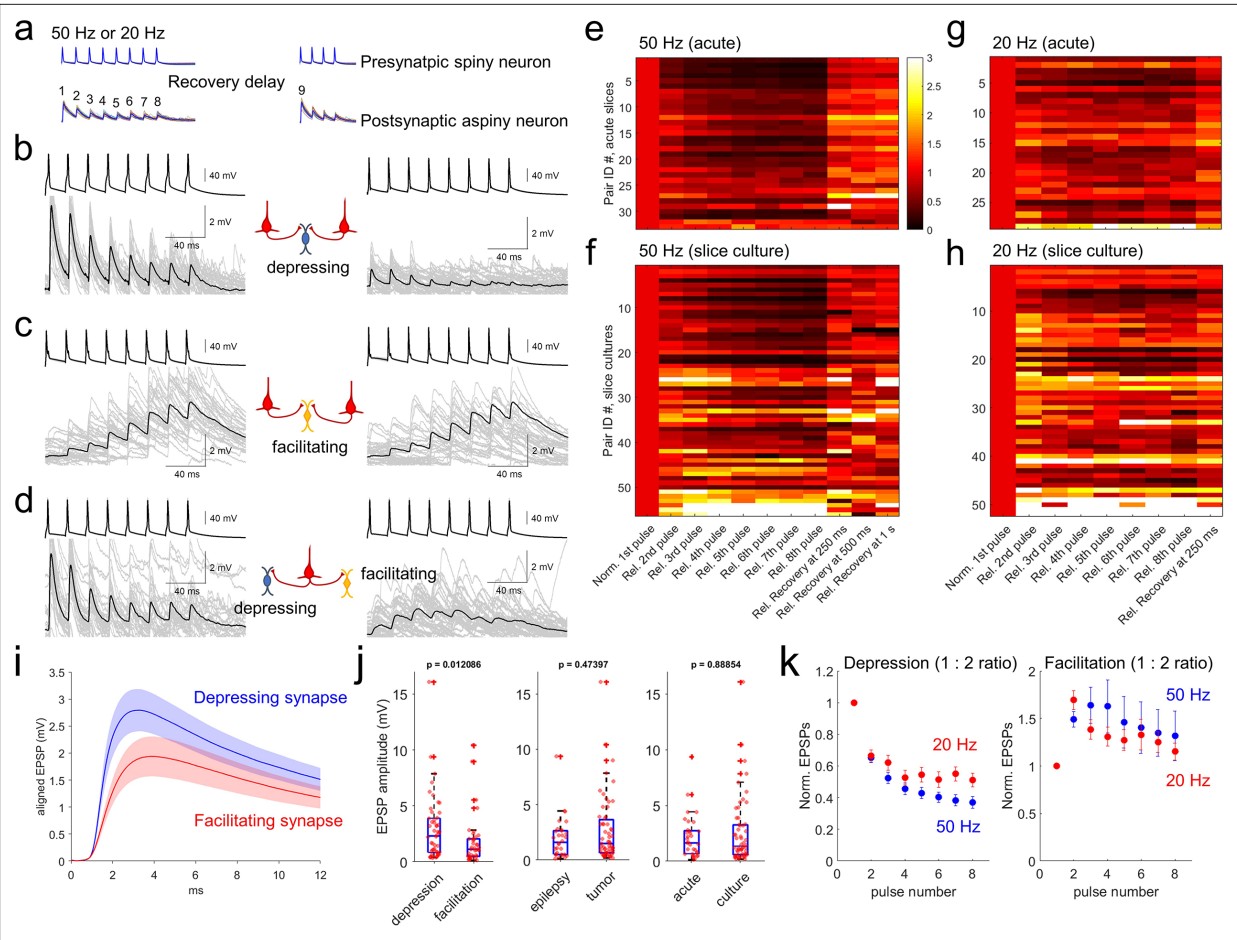

**Figure 4.** Synaptic dynamics of pyramidal-interneuron connections show heterogeneity and are target-cell specific. (**a**) Stimulation protocol of connectivity assays. Eight trained presynaptic spikes were generated with two main stimulus frequencies (20 and 50 Hz). Fixed 250ms recovery delay was used for 20 Hz stimulation and a range of recovery delays (from 62.5, 125, 250, 500ms, 1 s, 2, and 4 s) were interposed between the eight induction pulses and four recovery pulses. (**b–d**) Target cell-specific synaptic dynamics from pyramidal to interneuron connections. Two pyramidal neurons were connected to one interneuron and their synaptic dynamics were similar, that is, both were depressing (**b**), or both were facilitating (**c**). However, example in (**d**) shows one pyramidal neuron that was connected to two interneurons showing either depression (left panel) or facilitation (right panel). Averaged EPSP responses (blue, thick line) on top of individual responses traces (multiple colors) are displayed in each connected pair. (**e–h**) Initial EPSP sizes were normalized in connected pairs, and their relative synaptic dynamics according to presynaptic train stimulation (8 pulses) are displayed by heatmap. Heatmap rows sorted based on the size of EPSP, from largest (top row) to smallest (bottom row). Initial recovery pulse (denoted as 9 in panel **a**) responses out of 4 recovery pulses are displayed at 250ms, 500ms and 1 s at 50 Hz (**e,f**) and fixed recovery interval at 250ms at 20 Hz (**g,h**) as example responses. Synaptic connections found in acute slices (n=33 at 50 Hz in panel **e**); n=29 at 20 Hz in panel (**g**) and slice culture (n=56 at 50 Hz in panel **f**); n=52 at 20 Hz in panel (**h**). (**i**) Initial EPSP responses from 50 Hz stimulation were aligned from response onset and averaged. 1:2 ratio was determinant for classifying depression and facilitation at 50 Hz presynaptic stimulation. Aligned average EPSP kinetics are shown (depressing synapses, n=50; facilitating synapses, n=39). Displayed data indicate mean (blue, red)± s.e.m (shaded regions with light colors). (**j**) Amplitudes of EPSP responses (i.e. averaged first EPSP responses at 50 Hz stimulation in connected synapses) were compared between depressing (n=50) and facilitating (n=39) synapses defined by their 1:2 ratio at 50 Hz stimulation. EPSP amplitudes were compared from their tissue origins (n=30, epilepsy; n=59, tumor). EPSP amplitudes were also compared from their tissue preparation types (n=33, acute slice; n=56, slice culture). p-Values are from Wilcoxon rank sum test. (**k**) Kinetics of synaptic dynamics (1–8 pulses, normalized to first response) were compared at different frequencies (i.e. 50 Hz and 20 Hz presynaptic stimulation). Depression and facilitation synapses were defined based on 1:2 ratio. Kinetics of dynamics from depressing synapses are displayed (mean ± s.e.m; n=50, depression at 50 Hz, blue; n=29, depression at 20 Hz, red; left). Similarly, kinetics of dynamics from facilitating synapses are displayed (n=39, facilitation at 50 Hz; n=50, facilitation at 20 Hz; right).

The online version of this article includes the following figure supplement(s) for figure 4:

**Figure supplement 1.** Short-term synaptic dynamics with various methods (i.e. 20 Hz vs 50 Hz, and additional definition with 1:8 and 1:6–8 ratio) and impacts for disease condition or slice preparation.

**Figure supplement 2.** EPSP kinetic properties between depressing and facilitating synapses were not statistically different (Related to *Figure 4i*).

**Figure supplement 3.** Paired pulse ratios are strongly correlated with their recovery parameters.

*Figure 4 continued on next page*

*Figure 4 continued*

**Figure supplement 4.** No correlation found between paired pulse ratios and days after slice culture in synapse types (i.e., depressing and facilitating synapses).

of recordings (n=17) in slice cultures (total n=56) showed facilitation, compared to only 12% of recordings (n=4) in acute slices (total n=33; *Figure 4f and h*). This difference could either reflect an acute *vs.* slice culture difference, or more likely a selection bias for interneuron subtype sampling between slice preparation methods (see Discussion). To address the potential changes of short-term synaptic dynamics by increasing spontaneous synaptic activity that has been previously observed during slice culture (*Napoli and Obeid, 2016*), we analyzed the PPR change as a function of days in slice culture and did not observe a significant correlation (*Figure 4—figure supplement 4*).

These observed differences in pyramidal to interneuron synaptic properties could relate to target cell identity. Many differences have been described in pyramidal neuron to Pvalb-positive interneuron (depressing) and Sst-positive (facilitating) interneurons (*Reyes et al., 1998*; *Koester and Johnston, 2005*). In mouse V1, EPSP rise time and EPSP decay tau is shorter in pyramidal to Pvalb neurons compared to pyramidal to Sst neurons in mouse V1 (*Campagnola et al., 2021*). Perisomatically innervating Pvalb-positive basket cells allow rapid inhibition of neighboring neurons and shut down activity compared to dendritically innervating Sst-positive Martinotti cells (*Blackman et al., 2013*;

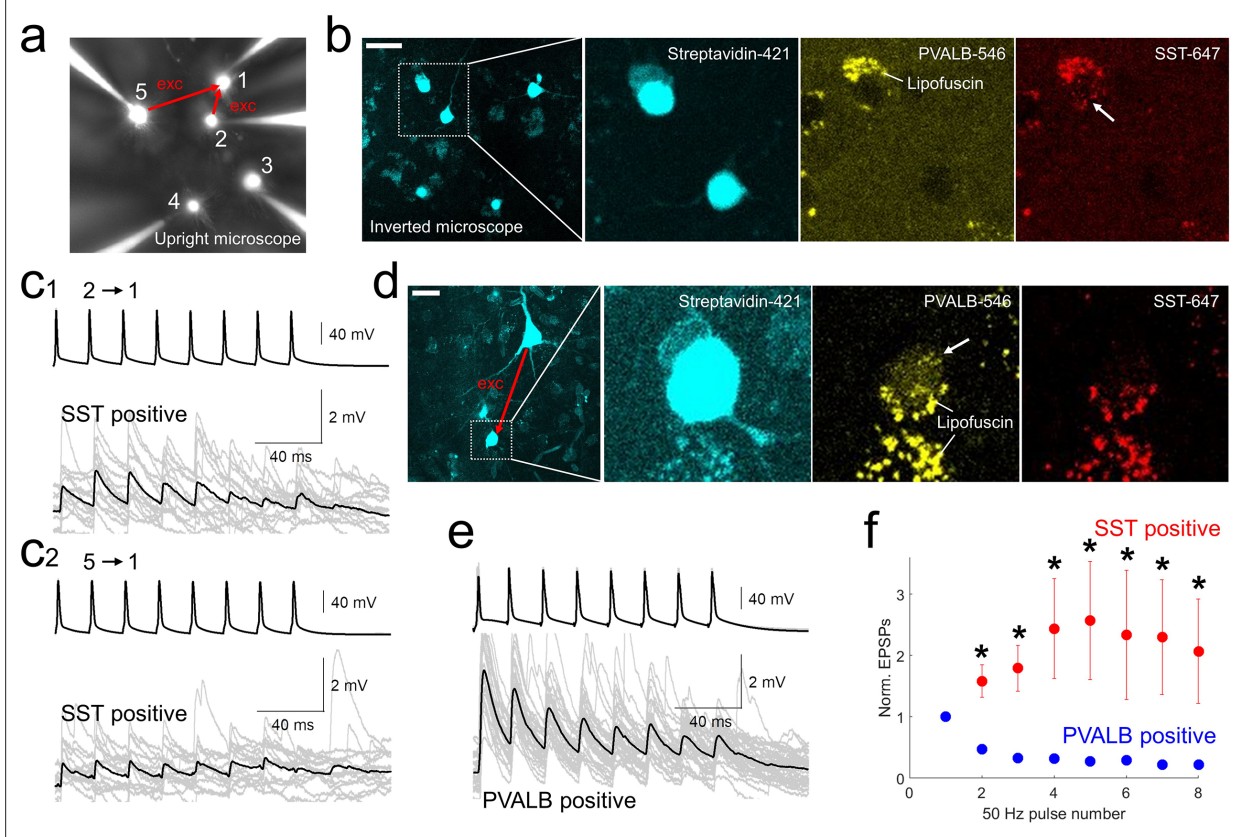

**Figure 5.** *post-hoc* HCR (mFISH) staining confirmed differentiated synaptic dynamics from excitatory pyramidal neurons to *PVALB* (depressing) and *SST* (facilitating) positive interneurons. (**a-b**).multiple patch-clamp recordings were performed underneath upright microscope (cascade blue, fluorescent dye included with biocytin on the patch pipettes, **a**) and biocytin filled patched cells, stained with streptavidin (left two panels, **b**) and their HCR staining were identified on the inverted microscope. Example of *SST*-stained neuron (cell 1) in connectivity assay from slice culture. Note that lipofuscin signal was seen in both PVALB (546 nm) and SST (647 nm) channels, but HCR signals were shown up separately in one or the other channel. **c1-2**, Corresponding synaptic dynamics at 50 Hz stimulation. In this example, two pyramidal neurons (cell 2 and cell 5) were connected to a SST-positive neuron (cell 1). (**d**) Example of *PVALB*-stained neuron in connectivity assay from acute slice (**d**). (**e**) Corresponding synaptic dynamics at 50 Hz presynaptic stimulation is displayed. Averaged EPSP responses (black, thick line) on top of individual responses traces (gray) are displayed for each connected pair in (**c1, c2**) and (**e**). (**f**) Summary plot of normalized synaptic dynamics from *PVALB* positive (blue; n=6) and SST positive (red; n=6) neurons. Displayed data indicate mean ± s.e.m (error bars; *, p<0.05 for both Wilcoxon rank sum test and false discovery rate (FDR) corrected Wilcoxon rank sum test).

*Lalanne et al., 2016*). Furthermore, frequency-dependent lateral inhibition between neighboring pyramidal neurons through facilitating Martinotti cells has been reported in both rodents (*Silberberg and Markram, 2007*; *Berger et al., 2009*) and human (*Obermayer et al., 2018*). Therefore, to directly investigate postsynaptic cell identity at the level of subclasses, we combined multiple patch-clamp recordings with either *post-hoc* HCR staining, or classifier-based predictions based on intrinsic membrane properties of postsynaptic interneurons in following sections.

## Cell subclass identification of postsynaptic interneurons by either mFISH staining or intrinsic membrane properties

Using *post-hoc* HCR, we were able to confirm the subclass identity of postsynaptic GABAergic cells that were connected to excitatory neurons (*Figure 5*). Initial EPSP amplitudes in the pairs with postsynaptic *PVALB*-positive neurons were larger compared to those of postsynaptic *SST*-positive neurons (EPSP amplitudes, mean ± s.e.m, 4.7393±1.3975 mV with *PVALB*-positive postsynaptic neuron, 0.8412±0.3147 mV with *SST*-positive postsynaptic neuron, p=0.0159, Wilcoxon rank sum test). Whereas synaptic dynamics with *PVALB*-positive neurons showed strong depression, synaptic dynamics with *SST*-positive neurons showed facilitation and their normalized trained responses were different (i.e. 2–8 pulse responses in *Figure 5f*; *, p<0.05, Wilcoxon rank sum test, n=5 for PVALB and n=6 for SST neurons).

Since *post-hoc* HCR on multiple patch-clamp recordings is a low-throughput method, we also took advantage of an existing human single-cell Patch-seq dataset to develop a quantitative classifier to predict interneuron subclass identity on our larger multiple patch-clamp recording dataset (*Lee et al., 2022*). This reference dataset comprised a set of Patch-seq experiments in slice culture that targeted AAV-DLX2.0-SYFP2 labeled neurons (*Berg et al., 2021*; *Lee et al., 2021*; see Methods), from which the cells were defined based on transcriptomic analysis following electrophysiological characterization. Such a classifier strategy was possible because intrinsic membrane properties of each cell were measured in our connectivity assays with multiple patch-clamp recordings, including subthreshold step hyperpolarization and depolarization from –70 mV holding potential and suprathreshold step depolarization (*Figure 3d, e, f and g*, and *Figure 6c*). To control for methodology-based differences between multiple patch-clamp recordings and single cell Patch-seq experiments, the two datasets were pre-aligned using supervised feature selection prior to classifier training and application (see Methods).

Using only these electrophysiological features, it was possible to differentiate between *PVALB*-positive GABAergic interneurons and other, non *PVALB*-positive interneurons, as illustrated in the UMAP in *Figure 6a*. The multiple patch-clamp recordings were intermingled with the Patch-seq neurons, indicating overlapping properties and coverage of both cell groups across the two datasets. A classifier trained on these intrinsic features from Patch-seq neurons predicted PVALB subclass identity with 76% accuracy (cross-validated prediction, with 29% false positive rate, 14% false negative rate). Importantly, quantitative predictions for the PVALB identity of postsynaptic interneurons from multiple patch-clamp recordings also matched well with mFISH labeling for those cells. Specifically, cells with positive *PVALB* labeling had high PVALB prediction probabilities, whereas cells with positive *SST* or *VIP* labeling had low PVALB prediction probabilities (*Figure 6b*). Examples of the intrinsic properties of a cell called as *PVALB*-positive by the classifier (with confirmed *PVALB* labeling), and a cell called as non-*PVALB* (and labeled positive for *SST*) are shown in *Figure 6c*.

The features with highest weighting in the classifier were AP height, depolarizing sag, AP upstroke adaptation ratio, and membrane time constant (*Supplementary file 2*). As shown in *Figure 6d*, these features were correlated with the classifier prediction of PVALB vs. non-PVALB identity. With this classification of postsynaptic interneurons measured in multiple patch-clamp recordings, we looked at the relationship between synaptic features and their PVALB probabilities (*Figure 6e*). As expected, cells with a high likelihood of being *PVALB*-positive tended to show synaptic depression as shown by the correlation between paired pulse ratios and PVALB probability using 50 Hz pulse trains. This result shows that paired pulse ratio is better metric (p=*0.0092262*) to predict PVALB type compared to EPSP amplitude (P=*0.079456*) (see also *Figure 6—figure supplement 1*). In addition, we looked at the relationship between EPSP amplitudes and paired pulse ratios within each type (i.e. PVALB vs non-PVALB) to see whether EPSP amplitudes have separable effects on short-term dynamics (*Figure 6—figure supplement 2*). The outcome shows that there is correlation between EPSP amplitudes and

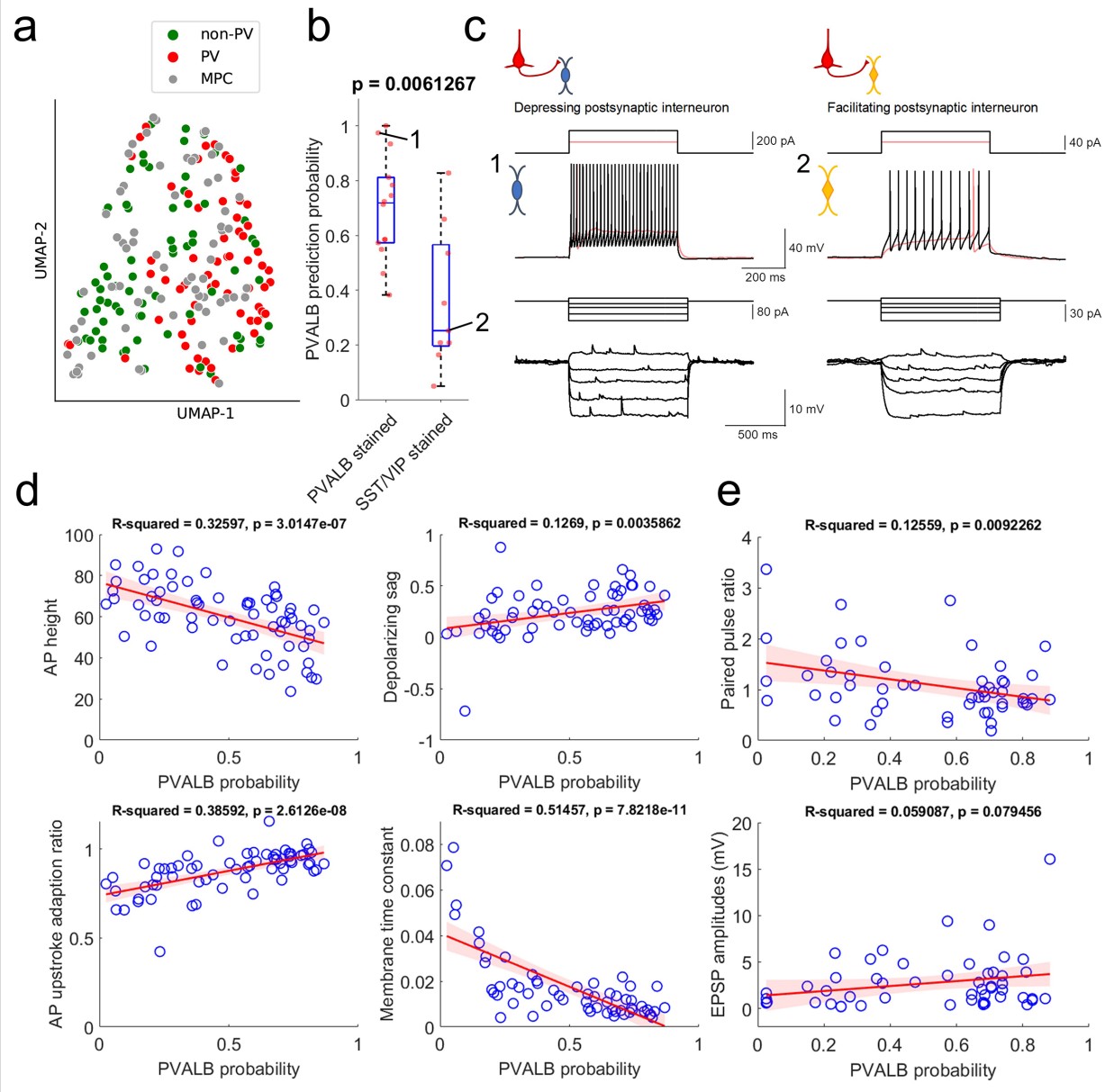

**Figure 6.** Prediction of PVALB and non-PVALB subclass cell type identities in postsynaptic interneurons by intrinsic membrane properties-based classifier using human Patch-seq data obtained from AAV virally labeled slice cultures. (**a**) UMAP visualization of PVALB and non-PVALB cell types based on their intrinsic membrane properties using human single cell Patch-seq data and alignment with postsynaptic cell intrinsic properties from multiple patch-clamp connectivity assay. (**b**) Correlation between PVALB probability predicted by intrinsic properties-based classifier and their subclass identity based on HCR staining (*PVALB* stained, n=14; *SST*/*VIP* stained, n=9). Classification based on PVALB and non-PVALB cell prediction. p Value from Wilcoxon rank sum test. (**c**) Example traces of intrinsic membrane properties from postsynaptic cells, showing synaptic depression (cell 1, *PVALB* positive) and facilitation (cell 2, *SST* positive). Two representative examples of outward step current injections (red and black trace) and corresponding voltage responses (red and black trace) shown in upper panels. (**d**) Examples of identified dominant features to segregate PVALB and non-PVALB interneurons. Regression line (red) with fitting confidence bounds (shaded region, light red) were displayed. (**e**) Correlation between PVALB probability and their paired pulse (1:2 ratio, upper) or EPSP amplitudes (lower) at 50 Hz presynaptic stimulation protocol. Regression line (red) with fitting confidence bounds (shaded region) in (**d**,**e**).

The online version of this article includes the following figure supplement(s) for figure 6:

**Figure supplement 1.** Classifier prediction with 20 Hz pulse trains.

**Figure supplement 2.** EPSP amplitudes and paired pulse ratios are correlated in non-PVALB neurons predicted by classifier.

**Figure supplement 3.** No correlation found between paired pulse ratios and days after slice culture in predicted subclass cell types (i.e. PVALB and non-PVALB).

paired pulse ratios especially in non-PVALB type, indicating that EPSP amplitudes and short-term dynamics may be related in a subclass cell-type specific manner. Again, as shown in *Figure 4—figure supplement 4*, we did not observe any significant correlation between PPR change as a function of days in slice culture when PVALB and non-PVALB neurons were used in the analysis (*Figure 6—figure supplement 3*).

## Discussion

### Target cell-dependent excitatory to inhibitory neuron synaptic properties in human cortex

Rodent studies have established that properties of short-term synaptic dynamics between excitatory and inhibitory neuron connections in the cortex and other regions are often dependent on post-synaptic neuron identity (*Blackman et al., 2013*). The current study establishes that this principle is also true in human cortex. Using multiple patch-clamp recordings in human neurosurgically resected cortical tissues, we find that individual pyramidal neurons show heterogeneous synaptic properties to multiple postsynaptic GABAergic neurons, and that those properties are defined by postsynaptic neuron identity. As in mouse, *Pvalb*-positive fast-spiking interneurons tended to show synaptic depression, whereas *Sst*-positive (or more generally, *Pvalb*-negative) interneurons tended to show synaptic facilitation (*Toledo-Rodriguez et al., 2005*). Given the conserved intrinsic properties of human and rodent PVALB neurons (i.e. fast spiking) and their target-specific depressing synaptic dynamics, *PVALB*-positive basket-like cells in human cortex are very likely to have similar functional roles in cortical circuits. These roles likely include mediation of excitation-inhibition balance, gain control, and generation/synchronization of fast oscillation (e.g. 'gamma' frequency range, 20–80 Hz) by communicating with reciprocally connected neighboring excitatory neurons (*Isaacson and Scanziani, 2011*). Non-PVALB interneurons, including *SST*-stained neurons by *post-hoc* HCR, instead showed rather small initial EPSP responses and tended to have short-term synaptic facilitation. These properties are comparable to previous studies in rodent *Sst*-positive 'Martinotti' cells, which are known to target pyramidal neuron apical dendritic tufts and mediate lateral disynaptic inhibition (*Silberberg and Markram, 2007*; *Berger et al., 2009*). Putative 'Martinotti' cells in human cortex also contribute to lateral disynaptic inhibition between two neighboring pyramidal neurons *via* receiving delayed facilitating synapses (*Obermayer et al., 2018*). Therefore, both the target cell-dependent principles and the specific properties of synaptic plasticity in PVALB versus other interneuron types appear to be strongly conserved, suggesting similar roles in functional cortical circuitry across species.

Transcriptomic analyses of interneuron subclass-selective gene expression suggest that synaptic properties may vary by interneuron subclass. Many synaptic genes are differentially expressed between PVALB and SST subclasses, and prior transcriptomic studies have suggested that such genes are particularly important for cell identity and function (*Paul et al., 2017*; *Huang and Paul, 2019*; *Smith et al., 2019*). While currently challenging to interpret at a gene-to-synaptic function level, since so many synapse-associated genes vary by cell subclass, eventually these molecular data could provide a mechanistic substrate for cell-type-specific functional properties and allow prediction of both conserved and divergent properties. For example, as mentioned above *Elfn1* has been shown to control short-term facilitation in SST interneurons (*Sylwestrak and Ghosh, 2012*; *de Wit and Ghosh, 2016*; *Stachniak et al., 2019*). We find that *ELFN1* is expressed in all interneuron subclasses except PVALB neurons, and this pattern is conserved from mouse to human, suggesting a similar role in non-*PVALB* expressing interneurons from mouse to human (*Jorstad et al., 2022*). Many other synapse-associated genes show differential expression across interneuron subclasses but with divergent expression across species (*Bakken et al., 2021*). While the functional significance of such differences remains to be demonstrated since there may also be multiple ways to achieve similar functional properties with different gene patterns of gene expression (*Goaillard and Marder, 2021*), several studies of human cortical tissues have shown functional differences between human and mouse. These include differences in excitatory neuron recovery from synaptic depression (*Testa-Silva et al., 2014*), higher presynaptic release probabilities (*Testa-Silva et al., 2014*), more docked vesicles (*Szegedi et al., 2016*), and polysynaptic network activities (*Molnár et al., 2008*; *Szegedi et al., 2017*; *Campagnola et al., 2021*), as well as diverse forms of synaptic plasticity among specific interneuron types

(*Verhoog et al., 2013*; *Szegedi et al., 2016*; *Mansvelder et al., 2019*). It seems likely that these functional synaptic species differences will ultimately be related to species variation in gene expression.

On the other hand, it is possible that physiological and synaptic properties may not be as discriminatory for cell specification as genes are, and that limited range and redundancy across types for these features. Patch-seq studies in mouse (*Gouwens et al., 2019*; *Gouwens et al., 2020*), monkey (*Bakken et al., 2021*), and human (*Berg et al., 2021*; *Lee et al., 2022*) suggest that there is a strong correlation of intrinsic and morphological features to highly granular transcriptomically defined cell types that would be averaged together at the 'subclass' level presented here. This may or may not be true at the level of synaptic physiology, but the tools are now available to begin addressing that question.

## Strategies for cell type-specific analysis of synaptic connectivity in human tissues

Directly analyzing synaptic properties of specific connected cell types by multiple patch-clamp recordings in human or other non-genetically tractable model organisms presents a number of challenges. The first challenge is simply access to healthy human tissues for slice physiology experiments. We and others have demonstrated that tissues from human neurosurgical resections are highly robust and can be used both for acute recordings and slice culture experiments over several weeks to months (*Eugène et al., 2014*; *Schwarz et al., 2017*; *Ting et al., 2018*; *Berg et al., 2021*). Another major challenge is efficiently targeting specific cell types. Typically, human tissue slice physiology is performed in unlabeled tissues with cell type targeting based only on their soma and proximal dendritic shapes under the microscope (*Molnár et al., 2008*; *Szegedi et al., 2016*). We have taken advantage of the longevity of human slices in slice culture to transduce neurons with enhancer-AAVs to allow viral transgenesis and genetic manipulation of cells in brain slices (*Andersson et al., 2016*; *Le Duigou et al., 2018*; *Ting et al., 2018*; *Mich et al., 2021*; *Schwarz et al., 2019*), in the current study to target GABAergic interneurons. Finally, another major challenge is the *post-hoc* identification of recorded neurons in multiple patch-clamp recordings. We demonstrate two effective strategies for cell type identification. The first is a low throughput but high confidence FISH staining of recorded neurons with markers of interneuron subclasses. The second is a quantitative classifier to differentiate interneuron subclass identities based solely on electrophysiology data, using a high confidence Patch-seq dataset that links physiology with transcriptomic identity to build the classifier. Together this array of solutions allowed the conclusions to be drawn about target cell-dependent synaptic properties at the GABAergic subclass level, and these approaches should be possible to apply at a much finer level of cell type resolution in the future.

A key strategy demonstrated here is to use mFISH with multiple rounds of staining on cleared thick in vitro human slice preparations, preserving tissue integrity and cell morphology, thereby allowing molecular identification of synaptically connected neurons using robust marker genes for neuron subclasses. The use of mFISH provides advantages over traditional immunohistochemical staining, as unambiguous identification of interneuron subclass identity (e.g. PVALB, SST, VIP, LAMP5) has not been reliable with *post-hoc* immunohistochemical staining in both non-human primate and human tissues. For example, PVALB antibodies work well (*Szegedi et al., 2017*), but SST and VIP antibodies do not work reliably in human cortical slices in our hands (data not shown; but see *Lukacs et al., 2022*). Here, the GABAergic interneuron subclasses PVALB, SST, and VIP were readily resolved using HCR and RNA transcript probes for *PVALB*, *SST*, *VIP,* and *LAMP5*. However, several challenges were identified for future improvement. Although mRNA labeling was robust for abundant genes, less abundant genes were more difficult to detect (including *PVALB* and *VIP*). Autofluorescence from lipofuscin, a common feature of human brain tissue, can complicate analysis and obscure mRNA signal. Improvement of lipofuscin mitigation techniques will facilitate future analysis. In some cases, we did not readily detect expected mRNA transcripts for cells with certain types of electrophysiological features (such as fast-spiking inhibitory neurons that would be expected to express *PVALB*). This could be the true state of the cell, or due to loss of mRNA through the patch pipette or leakage from the cell after pipette withdrawal in addition to HCR-based gene detection and amplification procedures in thick human surgical tissues. Finally, greater cell type resolution will be gained using more highly multiplexed mFISH techniques (*Chen et al., 2015*; *Eng et al., 2019*; *Wang et al., 2021*). The approach here constitutes a valuable step toward deciphering the correspondence of cell subclasses measured by multiple data modalities, despite the need for further technical refinement.

The development of AAV vectors for rapid infection and cell-type-specific transgene expression, provides new avenues for targeted analysis in the human brain as well as in non-genetically tractable organisms. This method provides a means to study neuronal and circuit properties in human neocortex and link them to emerging molecular classifications of cell types (*Tasic et al., 2018*; *Hodge et al., 2019*; *Bakken et al., 2021*). New enhancer-AAV tools now target wide variety of cortical cell types (*Graybuck et al., 2021*; *Mich et al., 2021*), and are promising new tools that could be applied to cultured organotypic slices to selectively mark or manipulate the cortical cells and circuits. However, two things that need to be carefully considered: one is potential culture artifact (*Ting et al., 2018*; *Suriano et al., 2021*), and the other is potential modification of synaptic properties in virally labeled neurons. Therefore, data obtained from virally transduced neurons in slice cultures ultimately need to be compared to data from both non-transduced neurons in slice cultures and acute slices (*Ting et al., 2018*; *Schwarz et al., 2019*). This difference we observed in this study, that is, more facilitating synapses were detected in slice cultures than in acute slices, could either reflect an acute *vs.* slice culture difference. However, we believe it is more likely to reflect a selection bias for PVALB neurons when patching in unlabeled acute slices, and that the AAV-based strategy with a pan-GABAergic enhancer allows a more unbiased sampling of interneuron subclasses whose properties are preserved in culture. In support of this, PPR analysis as a function of days after slice culture shows no relationship to acute versus slice culture preparation (*Figure 4—figure supplement 4*, *Figure 6—figure supplement 3*). Furthermore, we have observed that viral targeting of GABAergic interneurons greatly facilitates sampling of the SST subclass in the human cortex compared to unbiased patch-seq experiments (*Lee et al., 2022*), and this selection bias likely explains synapse type sampling differences in cultured slices compared to acute preparations. This ability to sample more representatively across GABAergic neurons is one of the advantages of the slice culture paradigm, and the rapidly increasing availability of new viral genetic tools, especially those validated to work in the context of human organotypic slices (*Qian et al., 2022*; *Mich et al., 2021*), should facilitate studies of human neuronal intrinsic properties and synaptic connectivity at increasing levels of cell-type specificity. Such viral tools also offer potential for cell-type-specific functional manipulation in mature human brain tissues.

Taken together, this combination of *post-hoc* marker labeling and computational classifier predictions indicate that we can identify postsynaptic cell identity for PVALB versus other, non-PVALB interneuron types in multiple patch-clamp recordings. With these postsynaptic cells identified, this allows a conclusion that synaptic properties between presynaptic human pyramidal neurons and postsynaptic interneurons are target-dependent based on the interneuron subclass identity, with PVALB neurons more likely to show synaptic depression and non-PVALB neurons more likely to show synaptic facilitation.

## Methods
### Transcriptomic analysis

Previously described single nucleus transcriptomic datasets from human MTG (*Hodge et al., 2019*) and mouse VISp (*Tasic et al., 2018*) were analyzed to define differentially expressed genes between PVALB and SST neuron types. Expression matrices were reduced to 14,870 orthologous genes conserved between human and mouse. A differential expression analysis between PVALB and SST subtypes was performed on log2 normalized data using the 'FindMarkers' function in Seurat v4.0.4 (*Hao et al., 2021*) with the Wilcoxon rank sum test. Genes were defined as differentially expressed if their log2 fold change was greater than 0.5 and their adjusted p-value was less than 0.01. Genes that were differentially expressed for PVALB or SST types in both species were used for heatmaps and gene ontology analysis (*Supplementary file 1*). For gene ontology analysis, the gene universe was defined by orthologous genes that had greater than 0 expression in PVALB or SST nuclei. The 'enrichGO' function from R package clusterProfiler (*Wu et al., 2021*; *Yu et al., 2012*) was used to compare conserved PVALB and SST DEG lists to the gene universe background with Benjamini-Hochberg correction and pvalueCutoff set to 0.01 and qvalueCutoff set to 0.05. Enriched terms were ranked by adjusted p-value and the top 10 terms for cellular compartment were shown.

p_val and p_val_adj indicate significance and adjusted significance of the differential expression test (Wilcoxon sum rank test). avg_log2FC means the average log2 fold change in expression between

the two cell populations (PVALB and SST). pct.1 is the proportion of target nuclei a gene is expressed in, and pct.2 the proportion of the background population the gene is expressed in.

## Acute slice preparation

Human cortical tissues were collected from adult patients undergoing neurosurgical procedures to treat symptoms associated with either epilepsy or brain tumor. Surgical specimens were obtained from local hospitals in collaboration with local neurosurgeons. All patients provided informed consent and experimental procedures were approved by Harborview Medical Center, Swedish Medical Center, and University of Washington Medical Center institute review boards before commencing the study. Surgically resected neocortical tissue was distal to the pathological core (i.e., tumor tissue or mesial temporal structures). Detailed histological assessment and using a curated panel of cellular marker antibodies indicated a lack of overt pathology in surgically resected cortical slices (*Berg et al., 2021*). In this study, we included data from 31 surgical cases, 15 of which were epilepsy cases and the remaining 16 were tumor cases (*Figure 2—figure supplement 1*). All specimens derived from neocortex with most cases derived from the temporal cortex (n=21) while a minority were obtained from the frontal cortex (n=9) or anterior cingulate cortex (n=1).

Surgical specimens were immediately transported (15–35 min) from the operating room to the laboratory in chilled (0–4°C) artificial cerebral spinal fluid (aCSF) slicing solution containing (in mM): 92 N-Methyl-D-glucamine (NMDG), 2.5 KCl, 1.25 $NaH_2PO_4$, 30 $NaHCO_3$, 20 4-(2-hydroxyethyl)–1-piperaz ineethanesulfonic acid (HEPES), 25 D-glucose, 2 thiourea, 5 Na-L-ascorbate, 3 Na-pyruvate, 0.5 $CaCl_2$, and 10 $MgSO_4$ (*Ting et al., 2018*). The NMDG aCSF was continuously bubbled with carbogen (95% $O_2$ and 5% $CO_2$). Osmolality was measured and adjusted to 300–315 mOsmoles/kg range (305–315 mOsmoles/kg range when using a freezing point osmometer, and 300–310 mOsmoles/kg range if using vapor pressure osmometer), and the pH was measured and adjusted to 7.3–7.4. 350-µm-thick human cortical slices were prepared using a Compressome VF-300 (Precisionary Instruments) or VT1200S (Leica Biosystems). After being cut, slices were transferred to oxygenated NMDG aCSF maintained at 34 °C for 10 min. Slices were kept at room temperature in oxygenated holding aCSF solution containing (in mM): 92 NaCl, 30 $NaHCO_3$, 25 D-Glucose, 20 HEPES, 5 Na-L-Ascorbate, 3 Na Pyruvate, 2.5 KCl, 2 $CaCl_2$, 2 $MgSO_4$, 2 Thiourea, 1.2 $NaH_2PO_4$ prior to recording (*Seeman et al., 2018*; *Berg et al., 2021*; *Lee et al., 2021*; *Campagnola et al., 2021*).

## Slice culture preparation

Following brain slice preparation and NMDG recovery steps as outlined above, a subset of brain slices were transferred to a six-well plate for culture and viral transduction. Human cortical brain slices were placed on membrane inserts (Millipore #PICMORG), and the wells were filled with 1 mL of culture medium consisting of 8.4 g/L MEM Eagle medium, 20% heat-inactivated horse serum, 30 mM HEPES, 13 mM D-glucose, 15 mM $NaHCO_3$, 1 mM ascorbic acid, 2 mM $MgSO_4$, 1 mM $CaCl_2$, 0.5 mM GlutaMAX-I, and 1 mg/L insulin (*Ting et al., 2018*). The slice culture medium was carefully adjusted to pH 7.2–7.3, osmolality of 300–310 mOsmoles/Kg by addition of pure $H_2O$, sterile-filtered and stored at 4 °C for up to 2 weeks. Culture plates were placed in a humidified 5% $CO_2$ incubator at 35 °C. 1–3 hours after brain slices were plated on cell culture inserts, brain slices were infected by direct application of concentrated AAV viral particles over the slice surface (*Ting et al., 2018*). The slice culture medium was replaced every 2–3 days until initiating synaptic physiology experiments. The time window to perform slice culture experiments ranged from 2.5 to 9 DIV, and a total of 36 cultured human neocortical slices were used in this study for the identification of gene expression with mFISH/HCR after multiple patch-clamp recordings.

## Viral vector production

Recombinant AAV vectors were produced by triple-transfection of ITR-containing enhancer plasmids along with AAV helper and rep/cap plasmids using the AAV293 cell line, followed by harvest, purification, and concentration of the viral particles. The AAV293 packaging cell line and plasmid supplying the helper function are available from a commercial source (Cell Biolabs). The PHP.eB capsid variant was generated by Dr. Viviana Gradinaru at the California Institute of Technology (*Chan et al., 2017*) and the DNA plasmid for AAV packaging is available from Addgene (plasmid#103005). Quality control

of the packaged AAV was determined by qPCR to determine viral titer (viral genomes/mL), and by Sanger sequencing of the AAV genome to confirm the identity of the viral vector that was packaged.

## CN1390 vector design and construction

Human neocortical interneurons were targeted in cultured slices by transducing slices with an optimized forebrain GABAergic viral vector CN1390, also known as pAAV-DLX2.0-SYFP2. The DLX 2.0 sequence includes a 3 x concatemer of the core region of a previously well-characterized DLX I56i forebrain GABAergic neuron enhancer (*Dimidschstein et al., 2016*; *Zerucha et al., 2000*). The 131 bp core sequence of the hI56i enhancer was inferred from enhancer bashing experiments detailed in *Zerucha et al., 2000*. The 393 bp 3 x core enhancer concatemer sequence was custom gene synthesized and subcloned into pAAV-minBetaGlobin-SYFP2-WPRE3-BGHpA upstream of the minimal promoter to make pAAV-DLX2.0-SYFP2, vector ID# CN1390 in our catalog. This vector will be deposited to Addgene for distribution to the academic community upon publication.

## Electrophysiology

Experiments were conducted on an upright microscope with an oblique condenser (WI-OBCD, Olympus) equipped with infrared (850 nm) illumination, 490 nm, 565 nm, and ultraviolet laser (395 nm) lines (Thorlab). 4 x and 40 x objectives (Olympus) were used to visualize the sample and a digital CMOS camera (Flash 4.0 V2, Hamamatsu) to take images. The rig configuration included eight electrodes disposed around the recording chamber, each surrounded by an headstage shield to prevent electrical crosstalk artifacts. Each patch electrode was positioned by x-y stage and micromanipulator (PatchStar, Scientifica) with guidance of acq4 open python platform software (http://acq4.org/; *Campagnola et al., 2014*). Bright-field and fluorescent images were also captured and analyzed with acq4. Signals were amplified using Multiclamp 700B amplifiers (Molecular Devices) and digitized at 50–200 kHz using ITC-1600 digitizers (Heka). Pipette pressure was controlled using electro-pneumatic pressure control valves (Proportion-Air; PA2193). The recording software, Igor Pro7 or 8 (WaveMetrics), contained with a custom software Multi-channel Igor Electrophysiology Suite (MIES; *Braun et al., 2022*), used to apply the bias current, inject the appropriate amount of current to patched cells, data acquisition and pressure regulation.

Slices were transferred to the recording chamber and perfused with carbogenated aCSF (2 mL/min), constant temperature (31–32°C), pH 7.2–7.3 and oxygen saturation in the recording chamber (40–50%). Perfusing aCSF contained (in mM): 1.3 $CaCl_2$, 12.5 D-Glucose, 1 or 2 $MgSO_4$, 1.25 $NaH_2PO_4$, 3 KCl, 18 $NaHCO_3$, 126 NaCl, 0.16 Na-L-Ascorbate. Patch pipettes were pulled from thick-wall filamented borosilicate glass (Sutter Instruments) using a DMZ Zeitz-Puller (Zeitz) to a tip resistance of 3–8 MΩ, and filled with intracellular solution containing (in mM) either 0.3 ethylene glycol-bis(b-aminoethyl ether)-N,N,N',N'-tetraacetic acid (EGTA) or no EGTA in addition to: 130 K-gluconate, 10 HEPES, 3 KCl, 0.23 $Na_2GTP$, 6.35 $Na_2$Phosphocreatine, 3.4 Mg-ATP, 13.4 Biocytin, and fluorescent dye with 50 µM Alexa-488 or cascade blue. Solution osmolarity ranged from 280 to 295 mOsmoles/kg titrated with sucrose, pH between 7.2 and 7.3 titrated with KOH. The liquid junction potentials were not corrected. For slice culture experiments, GABAergic neurons labeled with AAV-DLX2.0-SYFP2 were targeted during patch pipettes were approaching. With cascade blue loaded in the patch pipette, overlaid signals in the same cells with both SYFP2 and cascade blue were confirmed by manual inspection of image stacks with blue and green LED light excitation.

Cell cluster of eight neurons at each trial was selected and attempt for multiple patch-clamp recordings, targeted in mainly supraganular layer (L2 and L3), 50–80 µm depth from slice surface and smooth somatic appearance. Pairwise recordings were performed for local synaptic connectivity assay with both voltage and current-clamp mode. In voltage-clamp mode, membrane voltages of all patched cells were hold at either –70 or –55 mV and brief depolarization to 0 mV for 3ms at 20 Hz sequentially to reliably identify both excitatory and inhibitory connections. In current-clamp mode, initially all cell membrane potentials were maintained at –70±2 mV with automated bias current injection when we generated presynaptic unitary action potential by brief current injections (1.5–3ms) to detect EPSP responses in postsynaptic cells. For inhibitory connection, cell membrane potentials were maintained at –55±2 mV to detect IPSP responses in postsynaptic cells. Polysynaptic connections defined by postsynaptic long response latency were not considered in this study (see *Figure 4—figure supplement 2a*).

For the short-term plasticity, there are 12 action potentials at multiple frequencies (10, 20, 50, and 100 Hz) to induce sequential postsynaptic responses in connected pairs. Presynaptic stimulus amplitudes were adjusted to generate unitary action potential in each pulse. To measure recovery time course after induction protocol (i.e. initial 8 pulses), inter-spike interval between 8th and 9th pulses at 50 Hz stimulation was varied sequentially at 62.5, 125, 250, 500, 1000, 2000, and 4000ms. For other frequency stimulation (10, 20, and 100 Hz), we used fixed 250ms inter-spike interval between 8th and 9th pulses. Stimuli were interleaved between cells such that only one cell was spiking at a time, and no two cells were ever evoked to spike within 150ms of each other (*Seeman et al., 2018*; *Campagnola et al., 2021*). At each sequential 12 pulses stimulation for all patched neurons were repeated with 15 s inter-sweep interval. After running connectivity protocol, step current injections in each cell were applied to extract intrinsic membrane properties such as spike shape and frequency-current relationship.

## Human cortical interneuron patch-seq recordings in virally labeled slice cultures

Similar experimental procedures were applied as described in previous studies (*Berg et al., 2021*; *Lee et al., 2021*). Slices were bathed in warm (32–34°C) recording aCSF containing the following (in mM): (126 NaCl, 2.5 KCl, 1.25 NaH$_2$PO$_4$, 26 NaHCO$_3$, 12.5 D-glucose, 2 CaCl$_2$.4H$_2$O) and 2 MgSO$_4$.7H$_2$O (pH 7.3), continuously bubbled with 95% O$_2$ and 5% CO$_2$. The bath solution contained blockers of fast glutamatergic (1 mM kynurenic acid) and GABAergic synaptic transmission (0.1 mM picrotoxin).

Recording pipettes were filled with ~1.75 µL of RNAse Inhibitor containing internal solution: 110 mM K-Gluconate, 4 mM KCl, 10 mM HEPES, 1 mM adenosine 5'-triphosphate magnesium salt, 0.3 mM guanosine 5'-triphosphate sodium salt hydrate, 10 mM sodium phosphocreatine, 0.2 mM ethylene glycol-bis (2-aminoehtylether)-N,N,N',N'-tetraacetic acid, 20 µg/mL glycogen, 0.5 U/µL RNase Inhibitor, 0.5% biocytin, and either 50 µM Cascade Blue dye (excited at 490 nm), or 50 µM Alexa-488 (excited at 565 nm).

After examination of intrinsic membrane properties of virally labeled interneurons in conventional patch-clamp recordings, a small amount of negative pressure was applied (~–30 mbar) to extract the nucleus to the tip of the pipette. After extraction of the nucleus, the pipette containing internal solution, cytosol, and nucleus was removed from the pipette holder and contents were transferred into a PCR tube containing lysis buffer. cDNA amplification, library construction, and subsequent RNA-sequencing procedures are described in *Berg et al., 2021* and *Lee et al., 2021*. Patch-seq data was mapped to the reference taxonomy from human MTG dissociated cells (*Hodge et al., 2019*).

## Classification of intrinsic membrane properties in postsynaptic interneurons against a reference dataset obtained from single-cell patch-seq data

Intrinsic characterization of individual cells from both acute and slice cultures was carried out as described in *Campagnola et al., 2021*. Features were primarily calculated from sweeps with long square pulse current injection: subthreshold properties such as input resistance, sag, and rheobase; spike train properties such as f-I slope and adaptation index; and single spike properties such as upstroke-downstroke ratio, after-hyperpolarization, and width. For spike upstroke, downstroke, width, threshold, and inter-spike interval (ISI), 'adaptation ratio' features were calculated as a ratio of the spike features between the first and third spike. A subset of cells also had subthreshold frequency response characterized by a logarithmic chirp stimulus (sine wave with exponentially increasing frequency), for which the impedance profile was calculated and characterized by features including the peak frequency and peak ratio. Feature extraction was implemented using the IPFX python package (*Aitken et al., 2022*); custom code used for chirps and some high-level features will be released in a future version of IPFX.

Prediction of cell subclass from intrinsic properties was accomplished using a classifier trained against a reference dataset of cells with intrinsic properties and known subclasses. Reference cells were targeted in human slice culture with the same enhancer, AAV-DLX2.0-SYFP2 as the primary dataset, gene expression characterized using the patch-seq protocol (*Berg et al., 2021*; *Lee et al., 2021*), and transcriptomic subclasses assigned by mapping to a reference transcriptomic taxonomy from *Hodge et al., 2019* following the method from *Gouwens et al., 2019*. However, differences in

recording conditions between synaptic physiology and patch-seq protocols (primarily the presence/absence of synaptic blockers) cause shifts in intrinsic properties that preclude the use of all features in this reference dataset. We therefore excluded features for which the protocol accounted for over 5% of variance in an ANOVA of the combined dataset, leaving 24 features but excluding some common discriminating features such as spike width. Using scikit-learn, we trained a classifier pipeline that first normalizes features based on a robust variance (RobustScaler), imputes missing values based on nearest neighbors (KNNImputer), then classifies with linear discriminant analysis. This pipeline achieved 77% accuracy for predicting PVALB/non-PVALB subclasses in the reference dataset using cross-validation. Errors came primarily from a subset of SST cells with intrinsic properties overlapping the PVALB cells. The PVALB prediction probabilities of the classifier were then calibrated on the full reference dataset (CalibratedClassifierCV) before applying to the synaptic physiology dataset to generate both PV probabilities. Cells with a low confidence (PVALB probability between 0.4 and 0.6) were categorized as uncertain, with higher probabilities labeled PVALB and lower labeled non-PVALB. The separation of subclasses and overlap between patch-seq reference and connectivity assayed cells in intrinsic feature space was visualized using Uniform Manifold Approximation and Projection, UMAP (*Becht et al., 2018*).

## Data analysis

Synaptic connectivity and dynamics, intrinsic membrane properties were analyzed with custom-written MATLAB (MathWorks) and Igor (Wavemetrics) software. For statistical significance test, Wilcoxon rank sum test was used to compare two independent samples. Somatic position of individual neurons in a cluster from electrophysiological recording was imaged with fluorescent dyes (Alexa488 or cascade blue) with upright microscope and saved in ACQ4. Consequently, recorded neurons were identified with biocytin staining image and matched with mFISH/HCR signals taken by inverted confocal microscope.

To determine whether presynaptic spike generation is intact by a brief somatic current injection, all recorded presynaptic traces were manually checked and quality controlled based on the spike shape. When presynaptic spike shape is intact, postsynaptic response failures were included to average EPSP responses with multiple stimulations. EPSP onset delay was calculated from the peak of presynaptic spikes in current clamp mode to the onset of EPSP response. EPSP onset delay, PSP rise time, PSP decay tau were calculated with some modification of codes from Postsynaptic Potential Detector shared in public (MATLAB Central File Exchange; https://www.mathworks.com/matlabcentral/fileexchange/19380-postsynaptic-potential-detector, 2020).

## Thick tissue mFISH sample preparation

Slices were fixed in 4% PFA for 2 hr at room temperature (RT), washed three times in PBS for 10 min each, then transferred to 70% EtOH at 4 °C for a minimum of 12 hr, and up to 30 days. Slices were then incubated in 8% SDS in PBS at RT for 2 hr with agitation. The solution was exchanged with 2 X sodium chloride sodium citrate (SSC) three times, slices were washed for 1 hr at RT, followed by two additional (1 hr each) washes with fresh 2 X SSC.

## In situ HCR for thick tissue

We performed HCR v3.0 using reagents and a modified protocol from Molecular Technologies and Molecular Instruments (*Choi et al., 2014*). Slices were incubated in pre-warmed 30% probe hybridization buffer (30% formamide, 5 X sodium chloride sodium citrate (SSC), 9 mM citric acid pH 6.0, 0.1% Tween 20, 50 µg/mL heparin, 1 X Denhardt's solution, 10% dextran sulfate) at 37 °C for 5 min, then incubated overnight at 37 °C in hybridization buffer with the first three pairs of probes added at a concentration of 4 nM. The hybridization solution was exchanged 3 times with 30% probe wash buffer (30% formamide, 5 X SSC, 9 mM citric acid pH 6.0, 0.1% Tween 20, 50 µg/mL heparin) and slices were washed for one hour at 37 °C. Probe wash buffer was briefly exchanged with 2 X SSC, then amplification buffer (5 X SSC, 0.1% Tween 20, 10% dextran sulfate) for 5 min. Even and odd hairpins for each of the three genes were pooled and snap-cooled by heating to 95 °C for 90 s then cooling to RT for 30 min. The hairpins were then added to amplification buffer at a final concentration of 60 nM, and slices were incubated in amplification solution for 4 hr at RT. This was followed by a brief wash with 2 X SSC and a 1 hr, room temperature incubation in 2 X SSC containing 8 µg/µl Brilliant Violet 421TM

Streptavidin (BioLegend, Cat. No. 405225) and 0.05% Tween 20. Slices were washed three times for 10 min in 2 X SSC. For each round of imaging, an aliquot of 67% 2,2'-Thiodiethanol (TDE) solution was prepared for use as a clearing and immersion fluid. ≥99% TDE (Sigma-Aldrich) was mixed with DI water to create a 67% TDE solution with a refractive index of 1.46, verified by a pocket refractometer (PAL-RI, Atago). Slices were transferred to 67% TDE and allowed to equilibrate for at least 1 hr at room temperature prior to imaging.

## Quantification of thick tissue mFISH signals

Patched cells from acute and cultured tissues were hand segmented volumetrically using QuPath software (*Bankhead et al., 2017*; https://github.com/qupath/qupath, RRID:SCR_018257). Segmentation was performed on either the SYFP2 labeled cell body (slice culture preparation) or HCR signal (acute slice preparation) in transcript positive cells. Additionally, several nearby cells were also segmented in order to characterize typical expression levels in each probed gene and to compare signal level to patched cells. For each imaged channel, a histogram of non-cellular pixels was used to calculate a background threshold, which was taken to be three times the half width at half maximum above median of the distribution of pixel values. A mask of lipofuscin pixels was constructed by first taking all pixels that exceeded this threshold in all HCR channels. This mask was additionally expanded by morphological dilation with a kernel of radius one pixel, iterated two times. For each segmented cell, this mask was applied to each channel and the remaining intensity above background was integrated and normalized to the cell volume, this is taken as a measure of expression in each channel and reported in *Figure 3*, *Figure 3—figure supplements 1 and 2* and *Figure 5*.

## Confocal imaging

Thick tissue images were acquired on an Olympus FV3000 confocal microscope using a 30 X silicon oil objective with the zoom set to 1.5 x. The image montage stacks were acquired through the depth of the tissue at 1.2 µm steps. For figures, maximum intensity projections though the region of interest were generated are shown. Note that some montages exhibit stitching artifacts. Due to the frequent appearance of lipofuscin in aging human tissues, we showed HCR images as multiple overlapping channels since the lipofuscin granules were revealed as spots that are fluorescent in every channel.

## Stripping and subsequent hybridization rounds

To strip the signal in preparation for subsequent rounds, 67% TDE was exchanged with 2 X SSC three times and samples were washed for 1 hr. 2 X SSC was replaced with 1 X DNase buffer for 5 min and then a 1:50 dilution of DNase I in DNase buffer (DNase I recombinant, RNase-free, Roche, Cat. No. 04716728001), and incubated for 1 hr at 37 °C. This solution was replaced with fresh DNase solution before incubating slices overnight at 37 °C. Slices were washed with 65% formamide in 2 X SSC for one hour at 37 °C, then 2 X SSC for 1 hr at RT, before being transferred to 67% TDE for at least 1 hr. After imaging to confirm the signal was gone, the slices were washed in 2 X SSC for 1 hr to remove TDE before proceeding to subsequent hybridization rounds, which followed the protocol described above, except omitting the incubation in streptavidin solution.

## Morphological reconstruction

Reconstructions of the dendrites and the initial part of the axon (spiny neurons) and/or the full axon (aspiny/sparsely spiny neurons) were generated for a subset of neurons with good-quality electrophysiology and biocytin fills. Reconstructions were generated based on a 3D image stack taken by confocal microscope that was run through a Vaa3D-based image processing and reconstruction pipeline (*Peng et al., 2010*). The process could include a variable enhancement of the signal-to-noise ratio in the image (*Peng et al., 2014*). Reconstructions were manually corrected and curated using a range of tools (e.g., virtual finger, polyline) in the Mozak extension (Zoran Popovic, Center for Game Science, University of Washington) of Terafly tools (*Peng et al., 2014*; *Bria et al., 2016*) in Vaa3D. Every attempt was made to generate a completely connected neuronal structure while remaining faithful to image data. If axonal processes could not be traced back to the main structure of the neuron, they were left unconnected. As a final step in the manual correction and curation process, an alternative analyst checked for missed branches or inappropriate connections. Once the reconstruction was deemed complete, multiple plugins were used to prepare neurons for morphological analyses.

## Acknowledgements

We thank the Tissue Procurement, Tissue Processing, and Facilities teams for human tissue collection and brain slice preparation. We thank the hospital coordinators that help with logistics of collections and patient consent. We thank the Viral Technology team for packaging AAV vectors. We thank Lydia Potekhina and Shea Ransford for helping imaging on the confocal microscope. We thank Dr. Viviana Gradinaru for the gift of PHP.eB capsid packaging plasmid. We thank Dr. Christof Koch for comments on the manuscript and Dr. Gabe Murphy for leadership of the Synaptic physiology project. This work is supported in part by NIH BRAIN Initiative award RF1MH114126 from the National Institute of Mental Health to ESL, JTT, and BPL. The content is solely the responsibility of the authors and does not necessarily represent the views of the funding agencies. In addition, we wish to thank the Allen Institute for Brain Science founder, Paul G Allen, for his vision, encouragement and support.

## Additional information

### Competing interests

Jonathan T Ting, Boaz P Levi, Ed Lein: U.S. patent application #PCT_US2019_054539 related to this work (vector CN1390). The other authors declare that no competing interests exist.

### Funding

| Funder | Grant reference number | Author |
| --- | --- | --- |
| National Institutes of Health | BRAIN Initiative RF1MH114126 | Ed Lein<br>Boaz P Levi<br>Jonathan T Ting |

The funders had no role in study design, data collection and interpretation, or the decision to submit the work for publication.

### Author contributions

Mean-Hwan Kim, Conceptualization, Resources, Data curation, Formal analysis, Supervision, Validation, Investigation, Visualization, Methodology, Writing – original draft, Project administration, Writing – review and editing; Cristina Radaelli, Elliot R Thomsen, Deja Monet, Joseph T Mahoney, Investigation, Methodology; Thomas Chartrand, Resources, Data curation, Software, Formal analysis, Visualization, Methodology, Writing – review and editing; Nikolas L Jorstad, Resources, Data curation, Software, Formal analysis, Investigation, Methodology, Writing – original draft; Michael J Taormina, Software, Investigation, Visualization, Methodology; Brian Long, Software, Formal analysis, Investigation, Visualization, Methodology, Writing – original draft; Katherine Baker, Lindsay Ng, Jessica Trinh, Investigation; Trygve E Bakken, Hongkui Zeng, Supervision; Luke Campagnola, Resources, Software; Tamara Casper, Michael Clark, Nick Dee, Sara Kebede, Brian R Lee, Jim Berg, Kimberly A Smith, Tim Jarsky, Resources, Methodology; Florence D'Orazi, Project administration; Clare Gamlin, Staci A Sorensen, Resources, Visualization, Methodology; Brian E Kalmbach, Resources, Project administration; Charles Cobbs, Ryder P Gwinn, C Dirk Keene, Andrew L Ko, Jeffrey G Ojemann, Daniel L Silbergeld, Resources; Philip R Nicovich, Resources, Investigation, Visualization, Methodology; Jonathan T Ting, Conceptualization, Resources, Supervision, Methodology, Project administration, Writing – review and editing; Boaz P Levi, Conceptualization, Resources, Data curation, Formal analysis, Supervision, Methodology, Writing – original draft, Project administration, Writing – review and editing; Ed Lein, Supervision, Funding acquisition, Visualization, Project administration, Writing – review and editing

### Author ORCIDs

Mean-Hwan Kim ![ORCID] http://orcid.org/0000-0002-8065-4631
Thomas Chartrand ![ORCID] http://orcid.org/0000-0002-7093-8681
Joseph T Mahoney ![ORCID] http://orcid.org/0000-0003-1374-3893
Trygve E Bakken ![ORCID] http://orcid.org/0000-0003-3373-7386
Brian R Lee ![ORCID] http://orcid.org/0000-0002-3210-5638
C Dirk Keene ![ORCID] http://orcid.org/0000-0002-5291-1469
Tim Jarsky ![ORCID] http://orcid.org/0000-0002-4399-539X

Hongkui Zeng [ORCID] http://orcid.org/0000-0002-0326-5878

**Decision letter and Author response**
Decision letter https://doi.org/10.7554/eLife.81863.sa1
Author response https://doi.org/10.7554/eLife.81863.sa2

## Additional files

### Supplementary files

• Supplementary file 1. PVALB versus SST subclass differential gene expression analysis.

• Supplementary file 2. List of intrinsic membrane properties classifier features and linear discriminant analysis (LDA) coefficients.

• MDAR checklist

### Data availability

Single nucleus transcriptomic datasets from human MTG (*Hodge et al., 2019*) and mouse VISp (*Tasic et al., 2018*) are available in the Allen Institute website (https://portal.brain-map.org/atlases-and-data/rnaseq). Synaptic connectivity assay datasets including raw traces and related metadata information with MATLAB files (.mat), classifier analysis codes, and their intrinsic membrane property values are available in the DRYAD repository (doi:10.5061/dryad.jdfn2z3dm). Synaptic physiology experimental protocols and related topics are also available in the Allen Institute website (https://portal.brain-map.org/explore/connectivity/synaptic-physiology). To provide more publicly accessible data format, Neurodata Without Borders (NWB) files for synaptic connectivity assay performed in this study and human single cell patch-seq experimental data will be also available soon at DANDI or the BICCN data catalog.

The following dataset was generated:

| Author(s) | Year | Dataset title | Dataset URL | Database and Identifier |
|---|---|---|---|---|
| Kim MH, Radaelli C, Thomsen E, Monet D, Chartrand T, Jorstad N, Mahoney J, Taormina M, Long B, Baker K, Bakken T, Campagnola L, Casper T, Clark M, Dee N, D'Orazi F, Gamlin C, Kalmback B, Kebede S, Lee B, Ng L, Trinh J, Cobbs C, Gwinn R, Keene DC, Ko A, Ojemann J, Silbergeld D, Sorensen S, Berg J, Smith K, Nicovich P, Jarsky T, Zeng H, Ting J, Levi B, Lein E | 2023 | Target cell-specific synaptic dynamics of excitatory to inhibitory neuron connections in supragranular layers of human neocortex | https://doi.org/10.5061/dryad.jdfn2z3dm | Dryad Digital Repository, 10.5061/dryad.jdfn2z3dm |

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
