## [Editor Report]

The authors have made paired recordings from synaptically connected excitatory and inhibitory neurons in slices of human neocortex and used posthoc molecular methods to identify major subclasses of the recorded interneurons. The principal finding is that as found previously in the rodent cortex, the short-term plasticity of the synaptic connections from excitatory to inhibitory neurons depends on the molecular identity of the inhibitory neurons. Hence an important functional principle of connectivity is conserved.

---

## [Decision Letter]

**Decision letter after peer review:**

Thank you for submitting your article "Target cell-specific synaptic dynamics of excitatory to inhibitory neuron connections in supragranular layers of human neocortex" for consideration by *eLife*. Your article has been reviewed by 2 peer reviewers, and the evaluation has been overseen by a Reviewing Editor and Gary Westbrook as the Senior Editor. The following individual involved in review of your submission has agreed to reveal their identity: Sacha B Nelson (Reviewer #1). The reviewers have discussed their reviews with one another, and the Reviewing Editor has drafted this to help you prepare a revised submission.

Essential revisions:

The reviewers have made many suggestions for how to improve the manuscript to increase its clarity and accessibility. The most critical points concern:

1) The statistical analyses need to better address the core question: how much of the variance in synaptic properties depends on target identity vs. other factors.

2) The results chosen for presentation in the figures need to be better motivated, more clearly and accessibly described and the analyses need to be carried out and reported with greater rigor.

*Reviewer #1 (Recommendations for the authors):*

Because this manuscript reports on a large amount of hard works and very important data and makes an important point for the field, it is worth a bit more work to make the presentation clearer and more incisive in its ability to answer the posed questions.

The main issue is that it is very difficult to determine from the present analyses and the way these and the data are presented, the degree to which the observed differences in synaptic dynamics reflect postsynaptic identity vs. other factors.

Conceptually, it would be helpful to clearly state, up front the main variables being assessed: paired pulse depression vs. facilitation and Pvalb-positive vs. Pvalb-negative and then analyze what fraction of the variance in one is accounted by the other. Then one could go on to ask what fraction of the remaining variance is accounted for by other properties such as culture/acute and initial amplitude.

The remaining suggestions for improvement are organized sequentially according to the manuscript.

60: Kainite

The introduction might make the larger point that it is important to separate the biophysical properties which are conserved from those which are specialized for human and rodent since these will provide clues as to their contribution to conserved and specialized systems level functions.

Also, molecular differences between species may not all be functionally important and the same properties of connectivity and synaptic function might be achieved through distinct molecular mechanisms across species.

Figure 1:

The authors should do a better job of placing these results in the context of the massive amount of prior work on this issue, especially including that cited. The fact that this is reanalysis of existing data sets is stated in the methods, but should be stated in the results or figure legend for clarity. What is new here and what is confirmatory? Was the analysis of these ~75 markers done previously? How do these numbers of conserved markers compare to the number of conserved markers for other cell types?

The purpose of showing ELFN1 expression is quite clear, but the reason for focusing on the analyses in the rest of the figure could be better motivated/explained. Was the point about conservation of major type markers (Figure 1A) vs. poorer conservation of subtype specific markers (Figure 1-supp) made previously? If so, perhaps it does not not need to be made again. If there is a specific aspect of this point that has not been previously reported that needs to be clearly described.

Figure 3. The reason for including the phase plane analysis in panel g is unclear. Calling this "Quadruple modality data" is not especially helpful to alert the reader to what is being studied (and in fact it is a distraction to figure out which modalities are being considered separate). Instead it might be better to simply call this: example interneuron-pyramid connections in human cortex, or anatomy, physiology and molecular identification of interneuron-pyramid connections.

252: "In our MPC recordings, at least three cells were patched simultaneously, and we either simultaneously patched two presynaptic pyramidal neurons and one connected postsynaptic interneuron (Figure 4b,c), or one pyramidal neuron and two connected postsynaptic interneurons" This is confusing because presumably when larger numbers of neurons were patched there were more than two pre or postsynaptic neurons. Please indicate if you mean you are only considering cases in which there were precisely two, or whether you are also including cases if there were two or more (pre or post).

271: "whereas connected pairs with smaller EPSPs (lower heat map at 50 Hz; Figure 4e,f) tended to have facilitating synapses, suggesting that the large and small EPSP synapses may represent different inhibitory neuron types." This does not necessarily follow. There may be a spectrum across each type. This should be statistically addressed; either result: amplitude and short-term dynamics both follow type equally, or both type and amplitude have separable effects on short-ter dynamics, would be interesting.

Differences between acute and cultured slices: do the slice cultures develop any spontaneous activity? Short-term synaptic dynamics are strongly influenced by any ongoing activity.

294-308. This and the preceding paragraph deal with the question of what accounts for differences in short-term plasticity. The attempt to rule out slice and donor conditions is reasonable, but whether this depends only on subtype is still a question (at this point) and should be presented as such. The conclusion that subtype is the main or sole factor should be stated in the form: Therefore…depressing synapses=PVALB, facilitating = SST should come AFTER an analysis that directly addresses, e.g. using markers or analysis of presynaptic firing etc.

Figure 4 supp 1: please state frequency of this observation (relative to number of tests). Simply presenting an example is interesting, but not as helpful, especially given prior reports.

The results shown in Figure 4e-h show a clear continuum of mixtures of facilitation and depression. Even synapses that facilitate strongly during the first two responses can show strong depression later in the train. Hence classifying connections into facilitating and depressing is somewhat problematic. It is fine to make clear this is being done purely on the paired pulse responses but is this really a categorical difference? A slightly more nuanced way of talking about this would probably be appropriate.

Figure 4 supp 2. Given the results in Figure 4 supp 1, what criteria were used to judge that connections were monosynaptic, especially for those long responses with longer latency (e.g. >2 or 2.5 ms)

Figure 4. supp 3-I do not think this figure is especially helpful. A more correct way to analyze this is to parameterize the degree of facilitation/depression and simply analyze for a significant correlation between this parameter and the recovery parameter and use a two-way ANOVA with repeated measures to look for a relationship between recovery (the different times are repeated measures) and the slice and donor properties.

A more compact way to represent many of these data would be to fit a simple model like that used by Varela et al. (1997) or by Tsodyks and Markram (1997). Alternatively, perhaps it would be efficient to combine the data in Figure 4 supp 3 and Figure 4 supp 4-and perhaps only show a couple of comparisons (e.g. tumor-epilepsy and acute-culture). Since the data are made available, the figures should be focused on making specific points.

406: the use cell class specific

How often were slices stained for the markers shown in Figure 5 supp 1? The figure should report on all the data, not on a single slice or small number of slices. If this experiment was only done once or a few times this should be reported as such.

In some places the term "class" is used and in others "subclass" but these both appear to refer to major types of inhibitory neurons, not to subtypes of the Pvalb, SST and VIP classes defined by other markers.

479 a larger…datasets

Figure 6: The authors should use the available data to estimate a false positive and false negative rate for their classifier.

522-528. In addition to the possibility of a slice culture artifact or a bias in interneuron class, there may also be an acute injury artifact that recovers (or leads to cell death) after culturing. These could be addressed by asking (a) whether there is any difference in the incidence of non-Pvalb subtypes in culture vs. acute and (b) whether any other properties of cells or connections varied.

Discussion: the authors are quite bullish on the possibility of distinguishing many subtypes using the approaches used here to parse Pvalb from others, and are hopeful that these subtypes will correlate with other synaptic properties, but this rings a bit hollow in light of the fact that the electrophysiological classifier did not work as well in human cortex and the in situ results seemed so hard to obtain. In addition, it might be valuable to at least consider the possibility that the main properties of intrinsic and synaptic electrophysiology are specified much more coarsely then the fine gradations of marker expression, which may have other or even no function in the adult circuit.

*Reviewer #2 (Recommendations for the authors):*

1. My biggest concern with the series of big data paper coming out of the Allen Institute as of recently is that they are hard-to-understand descriptive data-dump papers. There's typically a massive amount of often unintelligible data with unclear implications. The authors really need to keep in mind what the point is of their paper is and how precisely the average reader will benefit from them. Just listing a bunch of abbreviations of genes (e.g. Figure 1 FS1) is generally not meaningful and only useful to a few people who are already in the know. All concepts must be properly introduced and clearly defined, otherwise this precious data will likely be somewhat lost on the community and the massive effort of the Allen Inst may be partially wasted. In this regard, I have tried to highlight particularly problematic issues, but I cannot possibly make an exhaustive list. I would recommend that the authors themselves try to identify problematic points and rectify them. Picture that the reader is a junior PhD student – how will they be able to understand the findings?

2. Brain region is often unclear, both in terms of results presented and literature discussed. For example, L49-62, text speaks of "Rodent studies" without specifying brain region, then proceeds to discuss Elfn1 (L57), which has only really been studied properly with OLM cells of the hippocampus (Ghosh papers cited on L58), but then on L61, it is somehow suggested that this "has also been reported in hippocampal pyramidal neurons", but that's where they were reported in the first place. This paragraph is so confusing that it is effectively misleading, because it seems to suggest that the literature mostly covers neocortex where the present Kim study was carried out, but that is not really the case. This is just an example, but is a problem that presents itself generally throughout this manuscript.

3. The paper often presents abbreviations as obvious, without proper definition or further explanation. While Pvalb, Sst, and Vip are reasonably worked-in concepts, I don't think e.g. LAMP5 (e.g. Figure 1d) or SLC17A7 (L442) are. Or SSv4 or Cv3 (Figure 1), or DLX (L203), or "GO: 0045211 term genes" (L133). But even if it were, proper definitions and explanations are necessary throughout. See Point 1 and e.g. Fig1FS1. Also, I thought the new abbreviations (as opposed to the old PV and SOM abbreviations) were Pvalb, not PVALB, Sst, not SST, etc. -- please clarify. The usage in the manuscript of PVALB vs Pvalb is furthermore not consistent (e.g. Figure 1d). We work with SWC files all the time (L1232), but is it useful for the broader readership to drop random abbreviations like that? And so on…

4. The statistical treatment is questionable at times. Throughout the manuscript, the authors talk about difference that are "significant" (e.g. L108, L176, L276, L287, L289, L291, 401, 414, 418, etc.). However, in science, differences are always significant, because if they're not significant, they're not differences. So no need to state that they're significant -- just state that they're different. On lines 298-308, the authors speak of "trends" that they think they see in the data. Please remove this entire discourse, as it is a misleading practice. Readers do not need to know what trends the authors think they see in their data. Differences are either there (as revealed by proper statistical treatment), or they are not. Recall that trends are real statistical concepts, e.g. you can in grouped data test for the significance of a trend using Cochran-Armitage's Test for Trend, so the authors shouldn't make claims about trends based on just looking at the data. L284-293, when the authors mix-and-match data using the eighth or the ninth pulse, do they properly account for multiple comparisons? Because these are not statistically independent measures, so if the same data set is used for e.g. eighth and ninth pulse, then a post-hoc adjustment must be carried out. Figure 4j, usage of inappropriate or poorly powered stats makes acute and cultured slices appear identical when they are not. Please use goodness-of-fit tests too, because it looks like although the means are indistinguishable by the rank test, the distributions are differently shaped. Numerical statistical methods may also help distinguish these differences. Or linear regression over the DIV using t-test for Pearson's r or non-parametric test for Spearman's rho. There are always differences between acute and cultured systems; the authors may want to make sure their study comes across rigorous in this regard.

5. Acute slice vs. cultured slice data should be better compared and differences discussed better. It is quite likely that many differences are simply artifacts due to the culturing approach. L279, L281-283: "More facilitating synapses were detected in slice cultures than in acute slices." and "This difference could either reflect an acute vs. slice culture difference, or more likely a selection bias for interneuron subtype sampling between slice preparation methods as discussed below." Or strong ones have undergone more presynaptic LTP? Cultured systems are usually spontaneously active, with bouts of high frequency firing. In fact, the difference in slice culture suggests that it is not a cell-type specific finding, but specific to the history of the cell or the synapse. (related to Point 4 above)

---

## [Author Response]

Essential revisions:The reviewers have made many suggestions for how to improve the manuscript to increase its clarity and accessibility. The most critical points concern:1) The statistical analyses need to better address the core question: how much of the variance in synaptic properties depends on target identity vs. other factors.

We would like to thank the editors and reviewers for distilling their comments to focus on this critical core question. We have now added additional statistical analyses and new figures (Figure 4 —figure supplement 3, Figure 4 —figure supplement 4, Figure 6 —figure supplement 2, and Figure 6 —figure supplement 3) in the revised manuscript as suggested. These additional analyses support the core result that synaptic properties depend on target cell identity rather than other factors such as disease condition or tissue origin.

2) The results chosen for presentation in the figures need to be better motivated, more clearly and accessibly described and the analyses need to be carried out and reported with greater rigor.

We have endeavored to improve the clarity of writing in the INTRODUCTION, in the Results section with newly added statistical analyses and addition/removal/rearrangement of figures, and the DISCUSSION. We hope the editors and reviewers agree this significantly improves the flow and makes the motivation for each figure clear. Hopefully this revised layout and additional analyses improves the rigor and accessibility of the study.

Reviewer #1 (Recommendations for the authors):Because this manuscript reports on a large amount of hard won and very important data and makes an important point for the field, it is worth a bit more work to make the presentation clearer and more incisive in its ability to answer the posed questions.The main issue is that it is very difficult to determine from the present analyses and the way these and the data are presented, the degree to which the observed differences in synaptic dynamics reflect postsynaptic identity vs. other factors.Conceptually, it would be helpful to clearly state, up front the main variables being assessed: paired pulse depression vs. facilitation and Pvalb-positive vs. Pvalb-negative and then analyze what fraction of the variance in one is accounted by the other. Then one could go on to ask what fraction of the remaining variance is accounted for by other properties such as culture/acute and initial amplitude.

We appreciate for the comments on this point and implementation of additional analysis and newly added figures follows the suggestion. We first show that synaptic dynamics with paired pulse stimuli show both depression and facilitation (Figure 4, and Figure 4 —figure supplements). Then, we used two methods to identify postsynaptic cells as PVALB-positive or PVALB-negative: post-hoc HCR (mFISH) (Figure 5), and a machine learning based classifier (Figure 6, and Figure 6 —figure supplement 1). Given the small number of postsynaptic cells that we were able to identify using the HCR staining technique, it was important to use a

classification-based grouping of PVALB-positive and PVALB-negative postsynaptic cell identities that allowed us to analyze a much larger dataset. These data showed differences in depression versus facilitation between synapses onto the PVALB-positive and -negative postsynaptic cell populations.

With this, we went on to ask what fraction of the remaining variance is accounted for by other variables. We analyzed the relationship between EPSP amplitudes and paired pulse ratio (Figure 6 —figure supplement 2) in each group and found that there is correlation between them especially in non-PVALB group but not in PVALB group which may indicate this relationship is subclass cell-type specific. We also asked if there was a systematic change in properties over time in culture by addressing the relationship between days after slice culture (DIV) and paired pulse ratio (Figure 6 —figure supplement 3, in addition to Figure 4 —figure supplement 4 for depressing and facilitating synapses). These analyses show that there is no correlation between them.

The remaining suggestions for improvement are organized sequentially according to the manuscript.60: Kainite

Thank you for pointing out this error. We have changed the text to “kainate receptors”.

The introduction might make the larger point that it is important to separate the biophysical properties which are conserved from those which are specialized for human and rodent since these will provide clues as to their contribution to conserved and specialized systems level functions.

Thank you for this suggestion. Our key motivation behind this study it to better understand how features vary between species. We have now added the following statement to the Introduction: “Thus, conservation of cellular properties between human and model organisms is often seen but cannot be assumed, and it is important to directly compare these properties to understand how well other organisms effectively model the human condition.”.

Also, molecular differences between species may not all be functionally important and the same properties of connectivity and synaptic function might be achieved through distinct molecular mechanisms across species.

We appreciate that molecular differences may not underlie functional differences, and that common functional properties may be achieved in a variety of different ways with different molecular expression of ion channels, receptors or other molecules as demonstrated so convincingly by Eve Marder’s work.

We now add a statement in the Introduction part, that "Whether these species differences lead to functional differences has been a topic of great debate, as it is well known that the same functional readouts, such as synaptic connectivity and dynamics, could be achieved through distinct molecular mechanisms across species (Goaillard and Marder, 2021).”. We also added a statement in the Discussion that effect, that “While the functional significance of such differences remains to be demonstrated since there may also be multiple ways to achieve similar functional properties with different gene patterns of gene expression (Goaillard and Marder, 2021), a number of studies of human cortical tissues have shown functional differences between human and mouse.”.

Figure 1:The authors should do a better job of placing these results in the context of the massive amount of prior work on this issue, especially including that cited. The fact that this is reanalysis of existing data sets is stated in the methods, but should be stated in the results or figure legend for clarity. What is new here and what is confirmatory? Was the analysis of these ~75 markers done previously? How do these numbers of conserved markers compare to the number of conserved markers for other cell types?

This specific comparison of PVALB vs SST directly is new and has not been done before, despite taking advantage of previously published datasets. It is this ability to support many different types of analyses that make the transcriptomic resources so valuable. In this study, we focused on the specific comparison of conserved genes across the species (i.e., mouse to human) that are differentially expressed between PVALB and SST interneuron subclasses (Supplementary Table 1). This specific analysis comparing across brain regions and different species (i.e., mouse visual cortex, VISp; mouse motor cortex, MOp; human primary motor cortex, M1; human middle temporal gyrus, MTG) is novel to our knowledge, and makes an important point about the likely importance of genes with strong conservation across diverse cortical areas and evolutionary time for the function of different neuronal types and their functional connectivity.

A comparison of the conservation of cell subclass markers was performed previously in Bakken et al. (2021), analyzing markers of each cortical GABAergic interneuron subclass versus all other GABAergic interneuron subclass combined. This showed that a core set of markers was robustly specific and conserved, although surprisingly most subclass-selective markers were not (now mentioned in the Results section). The current study focuses on the specific pairwise comparison of PVALB versus SST marker expression, and we feel a complete pairwise matrix comparison is out of scope for this particular study. Here, we focused on this smaller set of highly conserved markers of PVALB vs. SST that may plausibly underlie conserved structure and function across species.

The purpose of showing ELFN1 expression is quite clear, but the reason for focusing on the analyses in the rest of the figure could be better motivated/explained. Was the point about conservation of major type markers (Figure 1A) vs. poorer conservation of subtype specific markers (Figure 1-supp) made previously? If so, perhaps it does not need to be made again. If there is a specific aspect of this point that has not been previously reported that needs to be clearly described.

Thank you for highlighting our lack of clarity here. We have revised our Results section to clearly articulate the motivation behind this analysis. Original Figure 1 panel a,b,c were also rearranged to a,c,b and panel d was removed since it is less relevant for this specific point we were making.

Although this is a re-analysis of existing datasets, we wanted to better understand the expression of genes that were likely to control synaptic physiology. Since we are evaluating human SST and PVALB subclasses, that is where we focused these analyses. We consider the cross-species comparison of these two subclasses an important starting point for this study. The conserved expression of key physiological regulators of synaptic function (like ELFN1) across species suggests these molecules may be mediating conserved synaptic dynamics properties. Analysis for conserved PVALB and SST marker genes (Supplementary Table 1) were not done before and this suggests that there is a conserved core of genes controlling synaptic functions at the subclass level and accordingly, synaptic functional properties may be conserved across species. However, understanding the conserved and divergent elements of synaptic connectivity of other cell subclasses is beyond the scope of the work we are reporting.

Figure 3. The reason for including the phase plane analysis in panel g is unclear. Calling this "Quadruple modality data" is not especially helpful to alert the reader to what is being studied (and in fact it is a distraction to figure out which modalities are being considered separate). Instead it might be better to simply call this: example interneuron-pyramid connections in human cortex, or anatomy, physiology and molecular identification of interneuron-pyramid connections.

The reason for including the phase plane analysis in panel g is to compare spike shape of pyramidal neuron and fast spike interneuron. We added “and distinct spike shape compared to pyramidal neurons (Figure 3e,g)” in the text. We removed the term, “Quadruple modality data” throughout the manuscript and rewrote the Figure 3 title accordingly.

252: "In our MPC recordings, at least three cells were patched simultaneously, and we either simultaneously patched two presynaptic pyramidal neurons and one connected postsynaptic interneuron (Figure 4b,c), or one pyramidal neuron and two connected postsynaptic interneurons" This is confusing because presumably when larger numbers of neurons were patched there were more than two pre or postsynaptic neurons. Please indicate if you mean you are only considering cases in which there were precisely two, or whether you are also including cases if there were two or more (pre or post).

We did not mean we are only considering cases in which there were precisely two. We are also including cases if there were two or more (pre or post). We changed the text now to “In our multiple patch-clamp recordings, up to 8 neurons were targeted to patch simultaneously including both pyramidal neurons and interneurons. Therefore, we were able to include many recordings in our analysis such as two presynaptic pyramidal neurons and one connected postsynaptic interneuron (Figure 4b,c), or one pyramidal neuron and two connected postsynaptic interneurons (Figure 4d).”

271: "whereas connected pairs with smaller EPSPs (lower heat map at 50 Hz; Figure 4e,f) tended to have facilitating synapses, suggesting that the large and small EPSP synapses may represent different inhibitory neuron types." This does not necessarily follow. There may be a spectrum across each type. This should be statistically addressed; either result: amplitude and short-term dynamics both follow type equally, or both type and amplitude have separable effects on short-ter dynamics, would be interesting.

In Figure 4j (left), it shows that EPSP amplitudes are significantly bigger in the group of depressing synapses (p = 0.012086) compared to the group of facilitating synapses. However, as the reviewer 1 pointed out, these values may be a spectrum across each type and in fact, based on our classifier analysis with intrinsic properties (Figure 6e at 50 Hz), it shows that paired pulse ratio is better metric (p = 0.0092262) compared to EPSP amplitude (p = 0.079456) for PVALB cell type prediction. When we define two types as PVALB (as classifier probability over 0.6; n = 28) and non-PVALB (as classifier probability below 0.4; n = 20), short-term dynamics (i.e., paired pulse ratio) are statistically different in between these types (Wilcoxon rank sum test, p = 0.0208) but not in the case of EPSP amplitude (Wilcoxon rank sum test, p = 0.2079). Therefore, we rephrase to say that “This result shows that paired pulse ratio is better metric (p = 0.0092262) to predict PVALB type compared to EPSP amplitude (p = 0.079456).”.

In addition, we looked at the relationship between amplitude and paired pulse ratio within each type to see whether amplitude have separable effects on short-term dynamics (Figure 6 —figure supplement 2). The outcome shows that there is correlation between amplitude and paired pulse ratio especially in non-PVALB type but not in PVALB type, implicating that EPSP amplitude and short-term dynamics may have a correlation in subclass cell-type specific manner.

Differences between acute and cultured slices: do the slice cultures develop any spontaneous activity? Short-term synaptic dynamics are strongly influenced by any ongoing activity.

We didn’t systematically analyze the rate of spontaneous activity in the slice culture preparation, but we don’t typically observe spontaneous activity during these experiments. Consistent with our observed lack of spontaneous activity, a previous study has shown that robust spontaneous network activity emerges over longer periods of time in culture; nearly 40 days for human cultures and 10 days for mouse cultures (Napoli and Obeid, 2016; J of Cellular Biochemistry 117, 559-565). In this study, our human cultured slices were not used beyond 9 days, and we do not know the exact answer since we didn’t systematically measure it.

However, we have now performed additional analyses to assess the correlation between synaptic dynamics (1:2 paired pulse ratio in both depressing and facilitating synapses) and days of slice cultures (4 to 9 days after culture). We did not observe any correlation in both synapse types (i.e., depressing and facilitating synapses; Figure 4 —figure supplement 4) and subclass types (i.e., PVALB and nonPVALB; Figure 6 —figure supplement 3). Similarly, no correlation was observed with time in culture if we evaluated other metrics of synaptic dynamics such as 1:8 ratio,1-3:6-8 ratio. While these analyses do not directly address the question of changes in spontaneous activity over time in culture, at least, they add to the evidence that our measured synaptic properties are not affected by time in culture up to 9 days whether spontaneous activity changes over that time.

294-308. This and the preceding paragraph deal with the question of what accounts for differences in short-term plasticity. The attempt to rule out slice and donor conditions is reasonable, but whether this depends only on subtype is still a question (at this point) and should be presented as such. The conclusion that subtype is the main or sole factor should be stated in the form: Therefore…depressing synapses=PVALB, facilitating = SST should come AFTER an analysis that directly addresses, e.g. using markers or analysis of presynaptic firing etc.

We changed this paragraph accordingly and now the writing of this particular section of the RESULTS are rearranged and modified thoroughly, and it states in the end “These observed differences in pyramidal to interneuron synaptic properties could relate to target cell identity. Many differences have been described in pyramidal neuron to Pvalb-positive interneuron (depressing) and Sst-positive (facilitating) interneurons (Reyes et al. 1998; Koester and Johnston, 2005). In mouse V1, EPSP rise time and EPSP decay tau is shorter in pyramidal to Pvalb neurons compared to pyramidal to Sst neurons in mouse V1 (Campagnola, Seeman et al., 2022). Perisomatically innervating Pvalb-positive basket cells allow rapid inhibition of neighboring neurons and shut down activity compared to dendritically innervating Sst-positive Martinotti cells (Blackman et al., 2013; Lalanne et al., 2016). Furthermore, frequency dependent lateral inhibition between neighboring pyramidal neurons through facilitating Martinotti cells has been reported in both rodents (Silberberg and Markram, 2007; Berger et al., 2009) and human (Obermayer et al., 2018). Therefore, to directly investigate postsynaptic cell identity at the level of subclasses, we combined multiple patch-clamp recordings with either post-hoc HCR staining, or classifier-based predictions based on intrinsic membrane properties of postsynaptic interneurons in following sections.”.

Figure 4 supp 1: please state frequency of this observation (relative to number of tests). Simply presenting an example is interesting, but not as helpful, especially given prior reports.

We agreed and have decided to remove the figure from the manuscript.

The results shown in Figure 4e-h show a clear continuum of mixtures of facilitation and depression. Even synapses that facilitate strongly during the first two responses can show strong depression later in the train. Hence classifying connections into facilitating and depressing is somewhat problematic. It is fine to make clear this is being done purely on the paired pulse responses but is this really a categorical difference? A slightly more nuanced way of talking about this would probably be appropriate.

Thanks for the comments on this point and we agree that synapses that facilitate strongly during the first two responses often show strong depression later in the train, therefore paired pulse ratio (1:2 ratio) may only partially captures the properties of depression and facilitation. Therefore, we added an additional metric to define depression and facilitation with 1 to 6:8 ratio, and 1 to 8 ratio, and as a result, the property of activity dependent facilitation (i.e., 50 Hz compared to 20 Hz) in facilitating synapses, as known in mouse facilitating synapses, were slightly better captured compared to using paired pulse ratio (1:2 ratio) (Figure 4 —figure supplement 1a,b). In the Results, we stated as “Since PPR does not capture the full range of dynamics, we also defined additional metrics such as a 1:8 pulse ratio and a 1:6-8 pulse ratio defined by ratio between first and average of sixth to eighth pulses (Figure 4 —figure supplement 1a,b; Varela et al. Nelson, 1997; Tsodyks and Markram 1997; Beierlein et al., 2003).”.

Figure 4 supp 2. Given the results in Figure 4 supp 1, what criteria were used to judge that connections were monosynaptic, especially for those long responses with longer latency (e.g. >2 or 2.5 ms).

Although in our recent study (Campagnola et al., 2022), polysynaptic connections from human L2/3 pyramidal cells were inferred by response latency >3 ms versus PSP amplitude. However, in this study, polysynaptic connections defined by postsynaptic long response latency were not considered because it is only 2 data points over 3 ms onset delay as shown in Figure 4 —figure supplement 2a, and it mentioned now as “Polysynaptic connections defined by postsynaptic long response latency were not considered in this study (see Figure 4 —figure supplement 1a).” in the section of “Electrophysiology” of “Methods”.

Figure 4. supp 3-I do not think this figure is especially helpful. A more correct way to analyze this is to parameterize the degree of facilitation/depression and simply analyze for a significant correlation between this parameter and the recovery parameter and use a two-way ANOVA with repeated measures to look for a relationship between recovery (the different times are repeated measures) and the slice and donor properties.

There are two analyses implemented and added in the current manuscript. For the correlation between the paired pulse ratio and the recovery parameter, we define the recovery parameter as normalized mean values of early recovery time points (62.5 ms, 125 ms, and 250 ms) responses. (Figure 4 —figure supplement 3).

Second, two-way ANOVA test with repeated measures as different recovery time periods were performed and we didn’t observe any statistical significance (p > 0.05). These analyses indicate that there is no significant correlation between these parameters (the donor and slice properties) compared to overall recovery rates including each time points. Therefore, now Figure 4 —figure supplement 3 is removed and instead, we mentioned in the text accordingly: “Their recovery responses (i.e., 9^th^ pulse at various time intervals) were also not accounted for by disease indication (p = 0.4781) or slice preparation method (p = 0.7816, two-way ANOVA with repeated measures as different recovery time periods). Similarly, normalized synaptic dynamics (i.e., normalized responses from first to 8^th^ pulses at both 20 Hz and 50 Hz; Figure 4 —figure supplement 1c) were not impacted by disease state of the donor, or slice preparation method with two-way ANOVA with repeated measures (both p > 0.05).”.

A more compact way to represent many of these data would be to fit a simple model like that used by Varela et al. (1997) or by Tsodyks and Markram (1997). Alternatively, perhaps it would be efficient to combine the data in Figure 4 supp 3 and Figure 4 supp 4-and perhaps only show a couple of comparisons (e.g. tumor-epilepsy and acute-culture). Since the data are made available, the figures should be focused on making specific points.

We now removed Figure 4 – Supplement 3 and statistical analyses were implemented in the text. We hope this simplifies the descript data presentation. We didn’t perform any modeling on this study but given the relevance of these literatures on current our study, we now cited both references (Varela et al., 1997, Tsodyks and Markram, 1997) in the RESULTS.

406: the use cell class specific

Thank you for pointing this out and have tried to clean up the confusing language. We refer to excitatory and inhibitory as “cell classes”, and within the inhibitory cell class, PVALB, SST, LAMP5, and VIP constitute the main “subclasses”. The DLX2.0 AAV vector drives reporter expression in all the interneuron subclasses (including PVALB, SST, LAMP5 and VIP neurons) and thus is class-specific. Now we added in the Introduction, saying that “These neuron types are organized hierarchically, with levels referred to as class, subclass, and type. PVALB and SST neurons correspond to major divisions among GABAergic interneurons at the subclass level, along with LAMP5 and VIP subclasses.”.

How often were slices stained for the markers shown in Figure 5 supp 1? The figure should report on all the data, not on a single slice or small number of slices. If this experiment was only done once or a few times this should be reported as such.

We tested this specific combination of SLC17A7 and GAD1, and conducted this type of analysis in just the section shown in Figure 5 —figure supplement 1 a-d (now Figure 3 —figure supplement 1 a-d). However, staining of subclass interneuron marker genes including PVALB, SST, and VIP, as well as cell class markers SLC17A7 and GAD1, including co-staining with streptavidin was conducted dozens of times on dozens of unique tissues slices. We have updated the figure legend and the language in the text to clarify that we consider this a representative example of the staining we frequently observed.

In some places the term "class" is used and in others "subclass" but these both appear to refer to major types of inhibitory neurons, not to subtypes of the Pvalb, SST and VIP classes defined by other markers.

As mentioned above, consider excitatory neurons and inhibitory neurons as two different cell classes. We consider the SST, LAMP5, PVALB, and VIP populations of cortical inhibitory cells as subclasses. We now clarified in the Introduction, saying “These neuron types are organized hierarchically, with levels referred to as class, subclass, and type. PVALB and SST neurons correspond to major divisions among GABAergic interneurons at the subclass level, along with LAMP5 and VIP subclasses.”. That is that best resolution we can achieve with our current methods. There are many PVALB (7 types in MTG) or SST (11 types in MTG) cell types but we would require much higher resolution molecular profiling techniques to resolve them. We do not use the term cell subtypes and we have tried to stay consistent and be clear about how we use such terminology.

479 a larger…datasets

Thank you for pointing out this error. We have made “datasets” singular, and it now says “dataset”. We also now reference Lee et al. 2022 where this dataset is reported as “Since post-hoc HCR on multiple patch-clamp recordings is a low-throughput method, we also took advantage of an existing human single cell Patch-seq dataset to develop a quantitative classifier to predict interneuron subclass identity on our larger multiple patchclamp recording dataset (Lee et al., 2022).”.

Figure 6: The authors should use the available data to estimate a false positive and false negative rate for their classifier.

These performance details have now been added to the text: “A classifier trained on these intrinsic features from Patch-seq neurons predicted PVALB subclass identity with 76% accuracy (cross-validated prediction, with 29% false positive rate, 14% false negative rate).”

522-528. In addition to the possibility of a slice culture artifact or a bias in interneuron class, there may also be an acute injury artifact that recovers (or leads to cell death) after culturing. These could be addressed by asking (a) whether there is any difference in the incidence of non-Pvalb subtypes in culture vs. acute and (b) whether any other properties of cells or connections varied.

Thanks for the comments on this issue. For the question of (a), as we described in the RESULTS part, relatively “more facilitating synapses were detected in slice cultures than in acute slices. Based on the traininduced STP (1:6-8 ratio), about 30% of recordings (n = 17) in slice cultures (total n = 56) showed facilitation, compared to only 12% of recordings (n = 4) in acute slices (total n = 33)”. We agree this synapse types are not corresponding to PVALB and non-PVALB subtype. However, in our original manuscript we submitted, we described in the last part of the RESULTS part, “the percentage of neurons predicted to be non-PVALB neurons in acute slice recordings was much lower than in slice culture.”. In addition, as briefly described in the Supplementary Materials of the previous study (Campagnola et al., 2022), we were likely biased to record primarily fast-spiking PVALB neurons (based on targeting a small round shaped soma) in non-fluorescent labeled acute human slices. But viral labeling of GABAergic interneurons in cultured slices might allow us to access relatively less biased sampling of postsynaptic interneurons. This is our speculation based on the experiences. This idea is also consistent with the other recent study (Lee et al., 2022; https://www.biorxiv.org/content/10.1101/2022.11.08.515739v1) the authors showed that slice cultures transduced with the same virus we are using allowed a much higher incidence of sampling SST neurons than patching acute slices without the genetic reporter. However, it is still very hard to quantify cell-type specific injury artifact after culturing, but it is a good point, and we agree that we can’t exclude this possibility, such that SST neurons are likely more damaged during acute slice preparation and recover after culturing.

For the question of (b), paired pulse ratio as a metric of short-term synaptic plasticity didn’t change significantly along the days of slice culture as shown in newly implemented figures (Depressing and facilitating synapse, Figure 4 —figure supplement 4; PVALB and non-PVALB, Figure 6 —figure supplement 3) supporting the idea that synaptic properties such as facilitation and depression are likely conserved in each subclass group over the slice culture time window. For the change of intrinsic membrane properties are also described in Figure 1 of Lee et al., 2022. For example, action potential width and Sag currents were changed in PVALB WFDC2 neurons from the acute and culture paradigm.

Discussion: the authors are quite bullish on the possibility of distinguishing many subtypes using the approaches used here to parse Pvalb from others, and are hopeful that these subtypes will correlate with other synaptic properties, but this rings a bit hollow in light of the fact that the electrophysiological classifier did not work as well in human cortex and the in situ results seemed so hard to obtain. In addition, it might be valuable to at least consider the possibility that the main properties of intrinsic and synaptic electrophysiology are specified much more coarsely then the fine gradations of marker expression, which may have other or even no function in the adult circuit.

We think the comment “the fact that the electrophysiological classifier did not work as well in human cortex” seems to be referring to our original writing in L505 “Notably, we observed that the separation between PVALB-positive and other interneuron types is not as robust in human recordings compared to mouse”. That could depend significantly on sampling, so now we retracted this sentence. However, we think electrophysiological classifier did work relatively well with patch-seq data as shown in Figure 6 d,e.

We agree that the methods like mFISH that were applied in this study are challenging to execute. With the explosion of spatial transcriptomic methods, an improved and reliable gene detection method that works well in thick slices that have undergone physiological recording should be feasible for future studies. However, we expect this experimental paradigm to continue to generate significant challenges. In our opinion, better profiling techniques and increased throughput will be necessary to understand whether molecularly-defined subclasses and cell types are really predictive intrinsic and synaptic physiology.

We appreciate the viewpoint raised here based on the data at hand that intrinsic and synaptic properties may not have anything close to the fine specification that is apparent from transcriptomic (or even marker) specifications. While we view this as an open question to some degree, there is evidence of much greater stratification of physiological and morphological features that correlates with transcriptomically-defined cell types at a finer level of granularity than “subtype” such as Pvalb and Sst. Patch-seq efforts in mouse, monkey and human have shown this to be the case, although transcriptomically very similar types tend to have very similar physiology and anatomy. As such, the challenges of the intrinsic properties-based classifier to work here may be partly a result of lumping heterogeneous cell types together. On the other hand, it is also possible if not likely that other cellular phenotypes may simply not be very different across cell types, or sufficient to discriminate between types in the same way. At any rate, it is a question worth pursuing, and the current study begins to move in that direction with methods that in principle can be used to ask these questions at finer levels of cell type granularity. We have added a short paragraph on this topic in the Discussion, now shown in the last paragraph of first section “On the other hand, it is possible that physiological and synaptic properties may not be as discriminatory for cell specification as genes are, and that limited range and redundancy across types for these features. Patch-seq studies in mouse (Gouwen et al., 2019; Gouwen et al., 2020), monkey (Bakken et al., 2021), and human (Berg et al., 2021) suggest that there is a strong correlation of intrinsic and morphological features to highly granular transcriptomically defined cell types that would be averaged together at the “subclass” level presented here. This may or may not be true at the level of synaptic physiology, but the tools are now available to begin addressing that question.”.

Reviewer #2 (Recommendations for the authors):1. My biggest concern with the series of big data paper coming out of the Allen Institute as of recently is that they are hard-to-understand descriptive data-dump papers. There's typically a massive amount of often unintelligible data with unclear implications. The authors really need to keep in mind what the point is of their paper is and how precisely the average reader will benefit from them. Just listing a bunch of abbreviations of genes (e.g. Figure 1 FS1) is generally not meaningful and only useful to a few people who are already in the know. All concepts must be properly introduced and clearly defined, otherwise this precious data will likely be somewhat lost on the community and the massive effort of the Allen Inst may be partially wasted. In this regard, I have tried to highlight particularly problematic issues, but I cannot possibly make an exhaustive list. I would recommend that the authors themselves try to identify problematic points and rectify them. Picture that the reader is a junior PhD student – how will they be able to understand the findings?

We appreciate the reviewer’s comments and current version of manuscript was changed accordingly. We have made a serious effort to make this manuscript and its associated datasets more comprehensible to the average reader.

2. Brain region is often unclear, both in terms of results presented and literature discussed. For example, L49-62, text speaks of "Rodent studies" without specifying brain region, then proceeds to discuss Elfn1 (L57), which has only really been studied properly with OLM cells of the hippocampus (Ghosh papers cited on L58), but then on L61, it is somehow suggested that this "has also been reported in hippocampal pyramidal neurons", but that's where they were reported in the first place. This paragraph is so confusing that it is effectively misleading, because it seems to suggest that the literature mostly covers neocortex where the present Kim study was carried out, but that is not really the case. This is just an example, but is a problem that presents itself generally throughout this manuscript.

We rearranged this paragraph in the Introduction clearly as suggested as “Rodent studies from multiple brain regions have begun to elucidate differential synaptic properties between specific neuron types, as well as their underlying postsynaptic molecular mechanisms. For example, specific postsynaptic molecules controlling presynaptic transmitter release have been identified, including Elfn1 (extracellular leucine rich repeat and fibronectin Type III domain containing 1) (Sylwestrak and Ghosh, 2012), N-cadherin and β-catenin (Vitureira et al., 2012), PSD-95-neuroligin (Futai et al., 2007) in hippocampus, and Munc13-3 (Augustin et al., 2001) in cerebellum. In cerebral cortex, excitatory to morphologically defined multipolar basket cell synapses show a high initial release probability and synaptic depression. The GABAergic inhibitory interneuron basket cells are known to express the gene parvalbumin (PVALB); therefore, we will use the term PVALB interneurons to describe them, and use typical convention to refer to expression of the parvalbumin gene as PVALB for mRNA and PVALB for protein in human, and Pvalb for mRNA and PVALB for protein in rodent. In contrast, excitatory to morphologically defined bi-tufted (or low threshold activated, somatostatin-positive or SST interneurons) cell synapses show low initial release probabilities and synaptic facilitation (Reyes et al. 1998; Koester and Johnston, 2005). This specialized short-term facilitation in SST interneurons is known to be mediated by Elfn1 expression in postsynaptic dendritic shafts of SST cells (Sylwestrak and Ghosh, 2012; de Wit and Ghosh, 2016; Stachniak et al., 2019), but not in PVALB neurons. This molecular mechanism was originally discovered in the hippocampus but was extended to the cerebral cortex showing that Elfn1 in postsynaptic SST neurons interacts with presynaptic metabotropic glutamate receptors (mGluRs) and kainate receptors in a layer-specific manner (Stachniak et al., 2019).”.

3. The paper often presents abbreviations as obvious, without proper definition or further explanation. While Pvalb, Sst, and Vip are reasonably worked-in concepts, I don't think e.g. LAMP5 (e.g. Figure 1d) or SLC17A7 (L442) are. Or SSv4 or Cv3 (Figure 1), or DLX (L203), or "GO: 0045211 term genes" (L133). But even if it were, proper definitions and explanations are necessary throughout. See Point 1 and e.g. Fig1FS1. Also, I thought the new abbreviations (as opposed to the old PV and SOM abbreviations) were Pvalb, not PVALB, Sst, not SST, etc. -- please clarify. The usage in the manuscript of PVALB vs Pvalb is furthermore not consistent (e.g. Figure 1d). We work with SWC files all the time (L1232), but is it useful for the broader readership to drop random abbreviations like that? And so on…

We apologize for the confusing use of abbreviations. We have provided definitions of all abbreviations in current manuscript as below:

LAMP5 (e.g., Figure 1d, this plot is now removed), one of four major subclasses GABAergic interneurons, and we added in the text that “PVALB and SST neurons correspond to major divisions among GABAergic interneurons at the subclass level, along with LAMP5 and VIP subclasses.”.SLC17A7 and GAD1: We note that “Messenger RNA from prominent excitatory (SLC17A7, solute carrier family 17 member 17, also known as Vesicular Glutamate Transporter 1; Aihara et al., 2000) and inhibitory (GAD1, glutamic acid decarboxylase 1) marker genes were easily resolved in both patched (biocytin/streptavidin, StAv) and neighboring non-patched neurons (Figure 3 —figure supplement 1a-d).”.SSv4 and Cv3: We note in the figure caption (Figure 1a) saying that “SSv4 indicates SMARTseq V4 chemistry, and Cv3 indicates 10x Chromium V3 chemistry”.Gene ontology (GO): We explained in the text saying that “GO analysis can reveal if a gene set contains higher than expected number of genes associated with a cellular function or a subcellular compartment. The most significantly enriched GO terms were for synapse related categories, with postsynaptic membrane Gene Ontology term GO:0045211 (http://www.informatics.jax.org/vocab/gene_ontology/GO:0045211) being enriched in both PVALB and SST neurons.”.DLX2.0: It is close to the DLX5 and DLX6 genes and we denote in the text as “an optimized version of a previously described forebrain GABAergic neuron enhancer (Stuhmer et al., 2002; Dimidschstein et al., 2016; Mich et al., 2021).”.PVALB vs Pvalb: Capitals with italic indicate for Human genes and first letter with capital and rest letters with minuscules with italic indicate for mouse genes. In addition, non-italic with these letter rule indicates for protein, but not gene. We clarified this in the INTRODUCTION, saying that “The GABAergic inhibitory interneuron basket cells are known to express the gene parvalbumin (PVALB); therefore, we will use the term PVALB interneurons to describe them, and use typical convention to refer to expression of the parvalbumin gene as PVALB for mRNA and PVALB for protein in human, and Pvalb for mRNA and PVALB for protein in rodent.”.

To reduce the usage of abbreviation, we also removed “(saved as SWC files)” in the METHODS, Morphological reconstruction section.

4. The statistical treatment is questionable at times. Throughout the manuscript, the authors talk about difference that are "significant" (e.g. L108, L176, L276, L287, L289, L291, 401, 414, 418, etc.). However, in science, differences are always significant, because if they're not significant, they're not differences. So no need to state that they're significant -- just state that they're different. On lines 298-308, the authors speak of "trends" that they think they see in the data. Please remove this entire discourse, as it is a misleading practice. Readers do not need to know what trends the authors think they see in their data. Differences are either there (as revealed by proper statistical treatment), or they are not. Recall that trends are real statistical concepts, e.g. you can in grouped data test for the significance of a trend using Cochran-Armitage's Test for Trend, so the authors shouldn't make claims about trends based on just looking at the data. L284-293, when the authors mix-and-match data using the 8th or the 9th pulse, do they properly account for multiple comparisons? Because these are not statistically independent measures, so if the same data set is used for e.g. 8th and 9th pulse, then a post-hoc adjustment must be carried out. Figure 4j, usage of inappropriate or poorly powered stats makes acute and cultured slices appear identical when they are not. Please use goodness-of-fit tests too, because it looks like although the means are indistinguishable by the rank test, the distributions are differently shaped. Numerical statistical methods may also help distinguish these differences. Or linear regression over the DIV using t-test for Pearson's r or non-parametric test for Spearman's rho. There are always differences between acute and cultured systems; the authors may want to make sure their study comes across rigorous in this regard.

We changed “significantly different” to “different” or “statistically different” throughout the manuscript.

To quantify the trend whether EPSP amplitude and paired pulse ratio are correlated or not, we performed linear regression between paired pulse ratio and their EPSP amplitude, instead of Cochran-Armitage’s test, Author response image 1. However, based on the similar question from the reviewer 1, we did additional analyses whether this correlation is cell subclass-specific or not using the same dataset (i.e., PVALB and non-PVALB; Figure 6 —figure supplement 2).

**Author response image 1. sa2fig1:** 

For the concerns of mix-and-match data, two-way ANOVA with repeated measures were performed (as suggested from the reviewer 1). Furthermore, normalized values of individual pulse responses were pairwise compared for disease condition (tumor vs epilepsy) and tissue preparation (acute vs slice culture) (Figure 4 —figure supplement 1) with False discovery rate (FDR, Benjamini-Hochberg procedure) corrected Wilcoxon rank sum test and the statistical outcomes were described in the Results.In Figure 4j, in order to address whether the distributions are differently shaped, we performed additional statistical analysis with Kolmogorov-Smirnov test for a test decision for the null hypothesis that the data in two groups from the same continuous distribution. As a result, it shows the difference between “depression” vs “facilitation” (1:2 ratio) (left) p = 0.00029819, but not by “epilepsy” vs “tumor” (middle) p = 0.7852, and “acute” vs “culture” (right) p = 0.2766, which indicates data points were compared from the same continuous distribution with tissue origins (epilepsy vs tumor) or slice preparations (acute vs cultured).

Linear regression over the DIV in both PVALB and non-PVALB groups using t-test for Pearson’s r was performed in the newly added figure (Figure 6 —figure supplement 2).

5. Acute slice vs. cultured slice data should be better compared and differences discussed better. It is quite likely that many differences are simply artifacts due to the culturing approach. L279, L281-283: "More facilitating synapses were detected in slice cultures than in acute slices." and "This difference could either reflect an acute vs. slice culture difference, or more likely a selection bias for interneuron subtype sampling between slice preparation methods as discussed below." Or strong ones have undergone more presynaptic LTP? Cultured systems are usually spontaneously active, with bouts of high frequency firing. In fact, the difference in slice culture suggests that it is not a cell-type specific finding, but specific to the history of the cell or the synapse. (related to Point 4 above)

It was questioned from the reviewer 1 as well (shown below), and we did additional analyses and added the new supplementary figure (Figure 6 —figure supplement 2) accordingly.

One of the answers for the review 1 is repeated here:

Unfortunately, we didn’t systematically analyze the rate of spontaneous activity in the slice culture preparation, but we don’t typically observe spontaneous activity during these experiments. Consistent with our observed lack of spontaneous activity, a previous study has shown that robust spontaneous network activity emerges over longer periods of time in culture; nearly 40 days for human cultures and 10 days for mouse cultures (Napoli and Obeid, 2016; J of Cellular Biochemistry 117, 559-565).

Furthermore, we performed additional analysis for the correlation between synaptic dynamics (1:2 ratio in both depressing and facilitating synapses) along the days of slice cultures (4 to 9 days after culture) and we didn’t see any correlation, i.e., with paired pulse (1:2 ratio), r = 0.148, p = 0.305 for depressing synapses (n = 50 at 50 Hz), r = 0.161, p = 0.327 for facilitating synapses (n = 39 at 50 Hz). Similarly, in case we use different metric of synaptic dynamics such as 1:8 ratio,1-3:6-8 ratio, we didn’t see any correlation of synaptic dynamics metric as a function of progression of slice culture dates. Now we added this analysis in the Figure 4 —figure supplement 4. We performed the same analysis with PVALB and non-PVALB groups as mentioned above (Figure 6 —figure supplement 3) and we didn’t see any correlation.